# Influences of organic carbon speciation on hyporheic corridor biogeochemistry and microbial ecology

James C. Stegen [1], Tim Johnson[1], James K. Fredrickson[1], Michael J. Wilkins [2,3], Allan E. Konopka[1], William C. Nelson [1], Evan V. Arntzen[1], William B. Chrisler[1], Rosalie K. Chu [1], Sarah J. Fansler [1], Emily B. Graham [1], David W. Kennedy [1], Charles T. Resch[1], Malak Tfaily [1] & John Zachara[1]

The hyporheic corridor (HC) encompasses the river–groundwater continuum, where the mixing of groundwater (GW) with river water (RW) in the HC can stimulate biogeochemical activity. Here we propose a novel thermodynamic mechanism underlying this phenomenon and reveal broader impacts on dissolved organic carbon (DOC) and microbial ecology. We show that thermodynamically favorable DOC accumulates in GW despite lower DOC concentration, and that RW contains thermodynamically less-favorable DOC, but at higher concentrations. This indicates that GW DOC is protected from microbial oxidation by low total energy within the DOC pool, whereas RW DOC is protected by lower thermodynamic favorability of carbon species. We propose that GW–RW mixing overcomes these protections and stimulates respiration. Mixing models coupled with geophysical and molecular analyses further reveal tipping points in spatiotemporal dynamics of DOC and indicate important hydrology–biochemistry–microbial feedbacks. Previously unrecognized thermodynamic mechanisms regulated by GW-RW mixing may therefore strongly influence biogeochemical and microbial dynamics in riverine ecosystems.

[1] Pacific Northwest National Laboratory, Richland, WA 99352, USA. [2] Department of Microbiology The Ohio State University, Columbus, OH 43210, USA. [3] School of Earth Sciences, The Ohio State University, Columbus, OH 43210, USA. Correspondence and requests for materials should be addressed to J.C.S. (email: James.Stegen@pnnl.gov)

Groundwater–river water (GW–RW) mixing can stimulate dissolved organic carbon (DOC) turnover[1] in the hyporheic corridor (HC)[2], a critical domain of riverine ecosystems[3-6] responsible for up to 90% of ecosystem respiration[7]. A full accounting of the mechanisms that govern mixing-driven enhancements of biogeochemical activity and their ecosystem-scale consequences are prerequisites of process-based hydro-biogeochemical models that aim to predict the impacts of environmental change on watershed hydrology and biogeochemistry. In addition, a key challenge in working toward watershed-scale process-based hydro-biogeochemical models is to connect fine-scale processes to larger scale phenomena[8-10]. One mechanism through which mixing can enhance activity is by complementary electron donors (e.g., organic carbon) and terminal electron acceptors (e.g, oxygen) coinciding when GW mixes with RW, resulting in a biogeochemical hotspot[11,12]. By combining recent findings in marine and subsurface systems, we suggest an alternative, three-part mechanism: first, biologically labile DOC can accumulate along subsurface flow paths (as shown by Helton et al.[13]); second, concomitant decreases in DOC concentration protects this labile DOC (as shown by Arrieta et al.[14]); and third, in turn, GW–RW mixing combines low concentration labile DOC—protected from microbial oxidation by low total energy availability—with RW DOC that is higher concentration[1,15] but less labile[16].

We hypothesize that biogeochemical activity is, in turn, stimulated during GW–RW mixing due to GW-derived labile DOC 'priming'[17,18] the oxidation of poor quality river-derived DOC. Here we test this hypothesis—diagrammed conceptually in Fig. 1—from a thermodynamic perspective using ultra-high resolution profiling of DOC via Fourier transform ion cyclotron resonance–mass spectrometry (FTICR–MS). Upon finding support for the hypothesis, we examine the larger scale implications of mixing-driven enhancements for DOC biogeochemistry across

the HC by coupling end-member mixing models with time-lapse electrical resistivity tomography (ERT). We further reveal linkages between microbiome composition and DOC biochemistry, which highlight critical feedbacks among hydrology, DOC biochemistry, and microbial ecology.

## Results

**Mixing models and DOC thermodynamics.** We used water samples from GW wells and a near-shore piezometer to monitor subsurface aqueous chemistry through space (400 m spatial extent) and time (7-month temporal extent) (Fig. 2). Consistent with previous work[1], the analysis of water samples provided evidence that GW–RW mixing elevated heterotrophic respiration across the HC, but only within a narrow range of mixing conditions (~ 0–10% GW; Fig. 3). Per our conceptual model (Fig. 1), we hypothesized that intrusion of RW into the HC would stimulate microbial heterotrophic respiration[1,19,20], whereby DOC would decrease and dissolved inorganic carbon (DIC) would increase. An alternative hypothesis is that GW–RW mixing stimulates biogeochemical activity by bringing together complementary electron donors and terminal electron acceptors[11]. This mechanism is unlikely in the studied field system, because both end members (GW and RW) are oxygenated, whereby terminal electron acceptors would not be limiting. Nonetheless, to evaluate the potential for this mechanism, we took advantage of the higher concentration of DOC in RW and higher nitrate concentration in GW. In this case, the alternative hypothesis predicts that—in addition to a decrease in DOC and an increase in DIC, relative to a mixing model—GW–RW mixing will lead to losses of nitrate relative to a mixing model[12].

End-member mixing model results were consistent with our conceptual model. Measured DOC and DIC concentrations fell below (Fig. 3a, d) and above (Fig. 3b, e) a linear mixing model expectation, respectively; reactive solute concentrations (DOC, DIC, and nitrate) and conservative tracer concentrations (using $Cl^-$) in GW and RW were used to estimate a linear mixing model for each reactive solute[1] (see Methods). Consistent with stimulated heterotrophic respiration—and excluding purely abiotic processes such as sorption—the magnitudes of deviation were negatively correlated (Fig. 3f); greater losses of DOC were associated with greater increases in DIC, but only when considering conditions with less than ~ 10% GW (see purple boxes in Fig. 3d, e). However, nitrate did not deviate from the mixing model (Fig. 3c), thereby rejecting the alternative hypothesis that biogeochemical activity is stimulated by mixing complementary electron donors and acceptors.

Although elevated respiration under conditions of GW–RW mixing may be due to transport into the HC of labile DOC from RW and/or surficial sediments[1,19,20], our results are consistent with our conceptual model (Fig. 1). Our results also reject the hypothesis that labile DOC is derived from the RW. These inferences are supported by integrating DOC thermodynamics and concentrations with the mixing models. To examine DOC thermodynamics, we used FTICR–MS to assign molecular formulas to DOC species in our aqueous samples[21,22] and, in turn, calculated the Gibbs free energy of the half reaction of organic carbon oxidation[23] ($\Delta G^0_{Cox}$) (see Methods). Values of $\Delta G^0_{Cox}$ are positive such that DOC oxidation must be paired with the reduction of a favorable terminal electron acceptor[23].

A classic paradigm is that along subsurface flow paths microbes oxidize the most easily accessible DOC species, thereby leading to a decline in both DOC lability and concentration as residence time increases[20,24-26]. Based on this paradigm we hypothesized a negative relationship between $\Delta G^0_{Cox}$ and DOC concentration; $\Delta G^0_{Cox}$ values are positive and values closer to zero are more

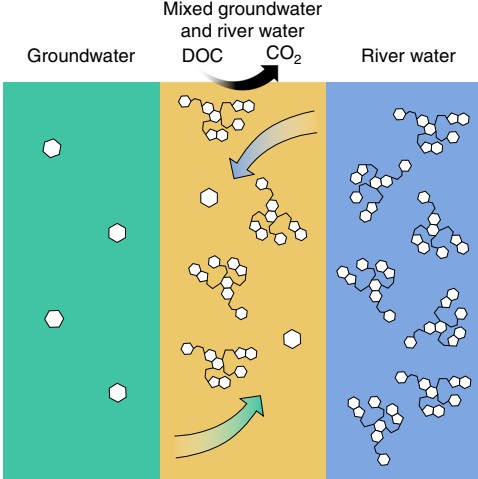

**Fig. 1** Hypothesized mechanism underlying the stimulation of respiration by mixing of groundwater with river water: low concentration of DOC protects thermodynamically favorable C (represented as monomeric C) by imposing an energetic limitation in terms of low total energy across the DOC pool, allowing favorable DOC to accumulate in GW. The thermodynamically unfavorable state of DOC (represented as polymeric C) in RW protects it from microbial oxidation, allowing it to accumulate in RW despite the DOC of RW being at a higher concentration. GW–RW mixing increases DOC concentration, relative to GW (removing the energetic protection), and improves DOC thermodynamic favorability (removing thermodynamic protection). Simultaneously removing these protection mechanisms stimulates aerobic respiration (DOC to $CO_2$)

In figure, labels: Groundwater | Mixed groundwater and river water — DOC → $CO_2$ | River water

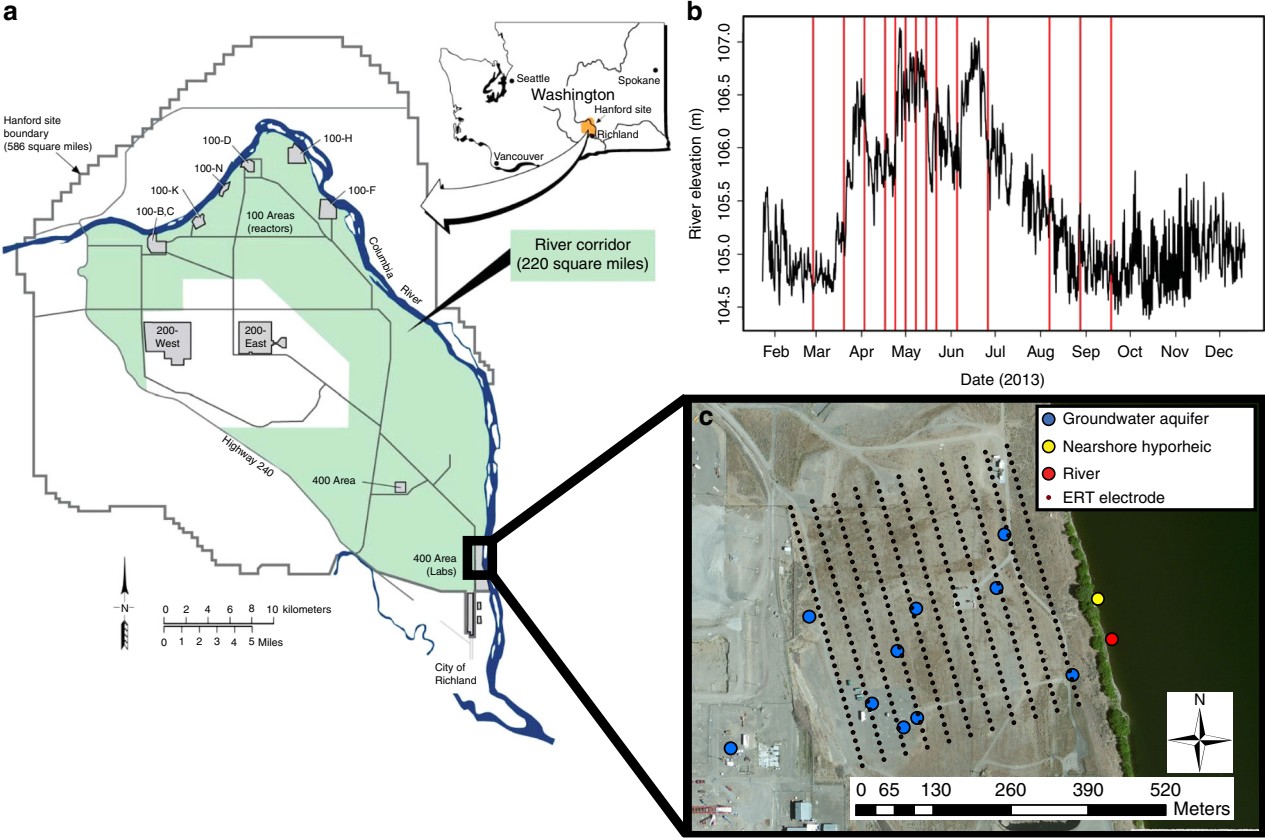

**Fig. 2** Spatial and temporal sampling design. **a** The location of the Hanford site within Washington State and the location of the Hanford 300 Area field site, reproduced from Stegen et al.[1]. **b** The Columbia River elevation near the field site throughout 2013; vertical red lines indicate sampling times. Sampling times targeted the most dynamic parts of the hydrograph and are not, therefore, evenly spaced. **c** Sampling locations within the field site where colors indicate different types of sampling locations; red indicates where water was sampled from the river water column, yellow indicates where nearshore hyporheic zone water was sampled from a piezometer placed 1 m below the riverbed, blue indicates where water was sampled from wells accessing the top of the groundwater aquifer, and small black circles indicate locations of geophysical electrodes; map was made using ArcMap 10.4

thermodynamically accessible for microbial respiration[23]. We further hypothesized that $\Delta G^0_{Cox}$ would increase with increasing GW fraction (i.e., DOC will be less thermodynamically favorable in GW due to consumption of more favorable DOC along flow paths[25]).

Our results directly conflict with the classic paradigm of decreasing DOC lability along subsurface flow paths, wherein we found that $\Delta G^0_{Cox}$ decreased with increasing GW fraction (Fig. 4a) and was significantly higher above a DOC concentration of 0.4 mg l$^{-1}$ (Fig. 4b). These results indicate that more thermodynamically favorable carbon accumulates in the GW where DOC concentrations are lowest (Fig. 3a), and that higher DOC concentrations in the river (Fig. 3a) are associated with less thermodynamically favorable DOC. Poor quality of DOC in RW is further supported by recent experiments showing that RW from the field system studied here does not elevate aerobic respiration relative to a DOC-free control[16]. Increased lability of organic carbon along longer flow paths is, however, consistent with recent work showing that carbon lability can increase with residence time in alluvial aquifers[13]. Those authors suggested that labile DOC—across long subsurface flow paths—may be derived indirectly through methanotrophy using methane that is generated in formations underlying the upper unconfined aquifer. Consistent with this explanation, DelVecchia et al.[27] showed that invertebrate biomass in the same aquifer is derived indirectly from methane. In our system, labile DOC in GW may also be indirectly derived from methanotrophy, as methane concentrations are elevated within fine-grained sediments that underlie the

coarse-grained sediments within which we sampled[28]. Labile DOC in the GW of our system may also be leached from buried organic carbon deposits (e.g., woody material), which occur sporadically in the aquifer[29].

Although labile DOC found in GW may be derived from a variety of sources, we propose that labile DOC is protected in the GW aquifer due to low DOC concentrations, as was recently shown for the deep sea[14]. In this case, labile DOC is protected by an energetic constraint, whereby the available DOC does not provide enough cumulative energy through time to each microbial cell, to offset the energetic costs of maintaining cellular machinery needed to oxidize the DOC[14]. We infer that this energy limitation is driven by very low GW DOC concentrations (~ 0.35 mg l$^{-1}$) that decrease the probability of a DOC molecule physically encountering a given microbial cell. In turn, DOC oxidation by microbial cells yields insufficient energy to offset maintenance costs (i.e., there is poor return on investment). This energetic constraint is based on the size of the DOC pool and contrasts with chemical constraints based on biochemical features of individual DOC species[30] and more physically based constraints such as mineral sorption or sequestration inside soil aggregates[31]. We further propose that the energetic constraint leads to rates of DOC oxidation that are too low to strongly influence the thermodynamic profile of the DOC pool. That is, we hypothesize that there is not enough oxidation of DOC to remove enough thermodynamically favorable compounds to drive the median $\Delta G^0_{Cox}$ toward higher values. In turn, low DOC

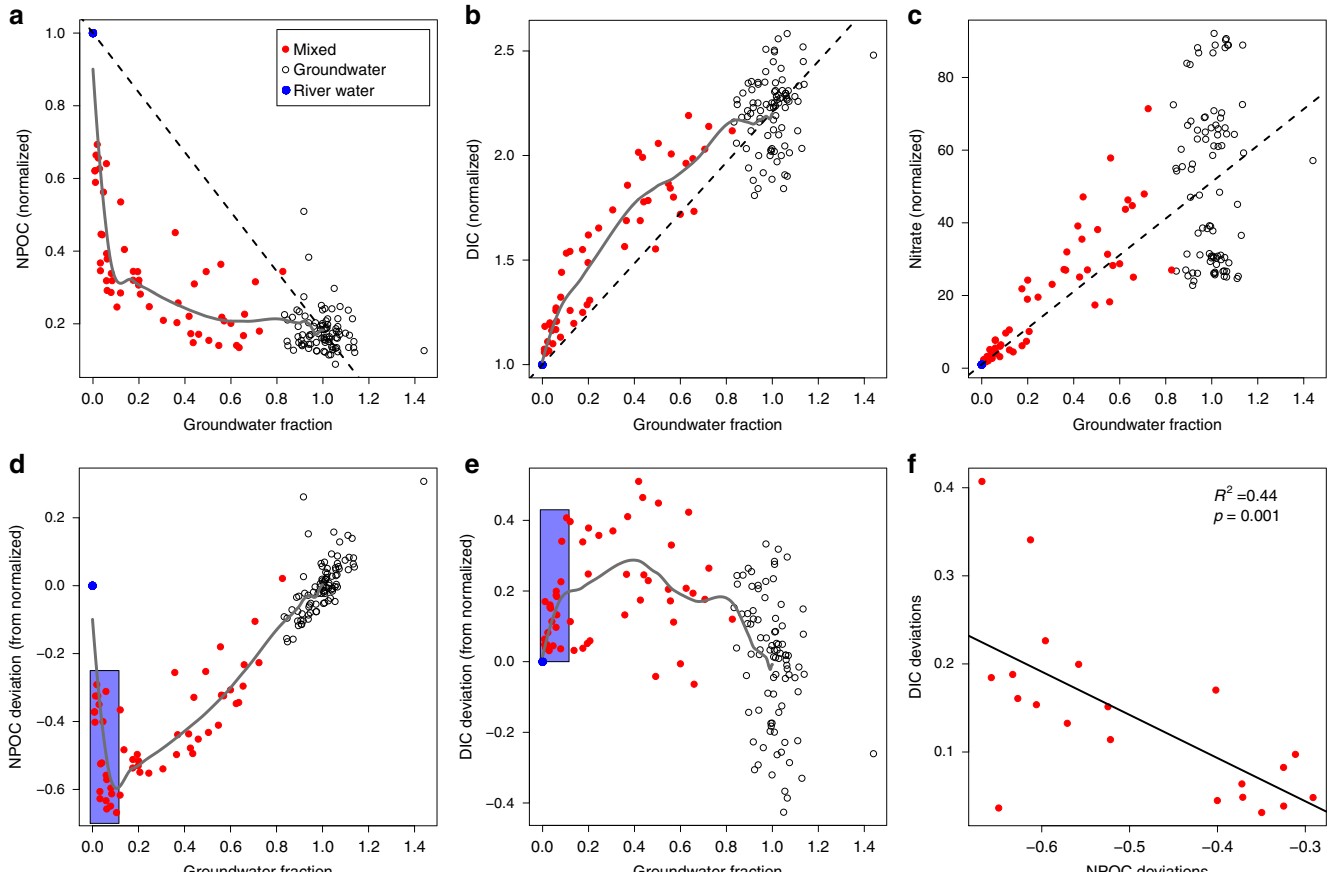

**Fig. 3** Reactive solutes and groundwater fraction and mixing model deviations. **a–c** Reactive solutes as functions of groundwater (GW) fraction inferred using Cl$^-$ as a conservative tracer. Linear mixing model expectations are indicated by dashed lines; gray lines are spline fits. River water (RW) concentrations of reactive solutes varied through time. Concentrations at each point in time were thus normalized to the associated RW concentrations; RW samples (blue circles) are always at 1 on the vertical axes. Cl$^-$ concentrations varied across GW wells (open circles) such that a threshold concentration was selected to indicate pure GW; non-river samples with Cl$^-$ concentration below this threshold were considered mixed (red circles). DOC, dissovled organic carbon; DIC, dissolved inorganic carbon. **d, e** Deviations from the DOC and DIC mixing models as functions of GW fraction. The purple rectangles on both panels indicate the range of conditions across which DOC deviations become increasingly negative. **f** DIC and DOC mixing model deviations within the purple rectangles are regressed on each other, revealing a significant negative relationship; the solid line is the regression model and statistics are provided on the panel

concentrations allow more thermodynamically favorable compounds to persist in the DOC pool.

Combining our DOC mixing model results with the observation of more labile DOC in GW supports the inference of an energetic constraint on DOC oxidation in GW. More specifically, along the continuum from pure GW to pure RW, there is a threshold at ~ 10% GW beyond which there is little change in DOC concentrations, which results in a linear trajectory—from 10% to 100% GW—toward the mixing model expectation (Fig. 3d). This suggests that once the DOC concentration decreases below some threshold, there is minimal oxidation of the remaining DOC regardless of how thermodynamically favorable the DOC is.

Identification of the DOC threshold below which there was little DOC oxidation aligned with the estimate of this threshold in the deep sea, suggesting a global constraint in saturated/aqueous environments. Preliminary analyses indicated that across the locations we sampled (Fig. 2) this threshold was at ~ 0.4 mg l$^{-1}$, which was confirmed as shown in Fig. 4b. Below this concentration, more labile DOC accumulated (although it does not always do so), and above this concentration DOC is dominated by less thermodynamically favorable species (Fig. 4b). Arrieta et al.[14] recently showed that DOC in the deep sea is

protected by low concentration, as opposed to being biochemically recalcitrant, which aligns with our inferences. They used controlled laboratory experiments to show that at DOC concentrations below ~ 0.35 mg l$^{-1}$ there is not enough energy to support metabolic machinery needed to oxidize DOC. Their lab experiment-based estimate of the DOC concentration threshold is very close to our estimate based on field observations. This correspondence suggests the intriguing possibility that DOC concentrations below ~ 0.35–0.4 mg l$^{-1}$ may globally constrain DOC oxidation in saturated/aqueous environments.

GW–RW mixing can stimulate biogeochemical activity by bringing together complementary electron donors and acceptors, leading to biogeochemical hot spots and moments[11,32]. Here we suggest an alternative mechanism, however, whereby GW supplies labile DOC that is energetically protected due to low concentration[14] and RW delivers DOC that is thermodynamically protected, whereby the mixing of these carbon pools overcomes their respective protection mechanisms. In this case, the GW-derived DOC can be thought of as 'priming'[17,18] the oxidation of river-derived DOC.

Although we can only speculate about mechanisms underlying this priming effect, we hypothesize that DOC thermodynamics may regulate expression and/or activity of enzymes used in the

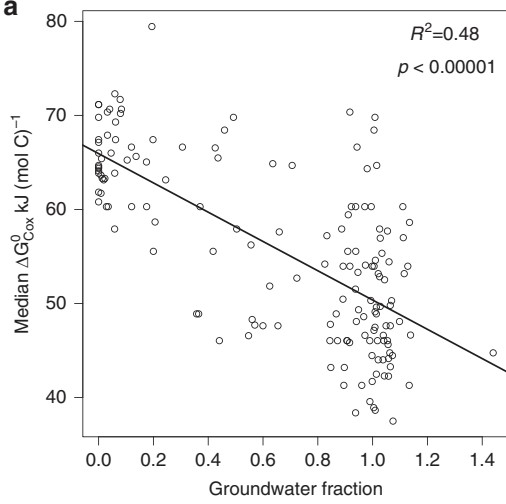

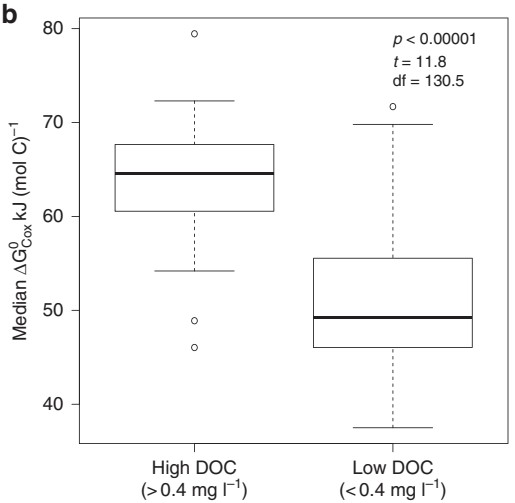

**Fig. 4** Patterns of within sample median Gibbs free energy of the organic carbon oxidation half reaction ($\Delta G^0_{Cox}$). **a** Median $\Delta G^0_{Cox}$ derived from FTICR–MS shows a significant negative relationship with the groundwater (GW) fraction, determined using Cl⁻ as a conservative tracer; thermodynamic favorability decreases with increases in $\Delta G^0_{Cox}$. Regression statistics are provided. **b** Boxplots summarizing the distributions of median $\Delta G^0_{Cox}$, which show a significant shift across a DOC concentration threshold of 0.4 mg l⁻¹; $t$-test statistics are provided. These two results indicate accumulation of thermodynamically favorable DOC in GW despite decreased DOC concentration along a gradient from river water to GW (Fig. 3a), thereby contradicting the classic paradigm of decreasing DOC quality along subsurface flowpaths

processing and oxidation of DOC. Under this hypothesis, inputs of low concentration and thermodynamically favorable DOC from GW act as a signal that results in increased enzyme expression or activity. The regulatory signal may be associated with the amount of energy gained per oxidation event. If the mechanism is via enhanced enzyme activity, it would suggest positive feedback regulation similar to that shown for the stringent response in *Escherichia coli*[33]. We further hypothesize that the thermodynamics-based signal only operates when DOC concentrations are above a threshold needed to offset the costs of producing enzymes used for DOC oxidation. As discussed above, this threshold appears to be ~ 0.35–0.4 mg l⁻¹ across both our HC system and the deep sea. Although speculative, this regulation-based hypothesis provides a starting point for future experiments designed to reveal underlying mechanisms.

**Spatiotemporal projections of DOC across the HC**. The patterns discussed above represent a largely unrecognized collection of processes that have the potential to strongly influence carbon cycling within the HC. Coupling the mixing models with time-lapse ERT[34–36] further indicates that these processes have a strong influence over HC-scale DOC dynamics (see Methods and Supplementary Figs. 1–3). In particular, processes leading to the nonlinear relationship between DOC and GW fraction—due, in part, to processes captured in Fig. 1—result in thresholding behavior in the spatial distribution of DOC. Specifically, across the HC significant increases in DOC were projected to occur only when the GW fraction dropped below 10% (Fig. 5). Leveraging the contrast in electrical conductivity (EC) between GW and RW in the study system, ERT[34] imaging was used to characterize the intrusion of RW into the HC (Fig. 5a, d). RW pulsed in and out of the HC during our 7-month study period due to a combination of seasonal- and dam-controlled variation in river stage (Fig. 2)[34]. This resulted in a broad range of GW–RW mixing conditions across the HC. Over this period, ERT was used to produce images of changes in aquifer EC resulting from RW intrusion and retreat, which were subsequently converted to spatiotemporal estimates of GW fraction. The statistical relationship between DOC and GW fraction (gray line in Fig. 3a) was coupled to the ERT images, in order to project DOC concentrations across the ERT domain at each point in time.

Coupling the aqueous chemistry to the ERT images revealed that large intrusion events that lowered the GW fraction in the HC to ~ 15% (Fig. 5a) had little effect on the DOC concentration (Fig. 5b) due to substantial losses of DOC during RW intrusion (Fig. 5c). ERT projections further highlight an important impact of the nonlinear relationship between DOC and GW fraction; a relatively small decrease in the GW fraction from ~ 15% to 10% (Fig. 5d) may lead to significantly more DOC being transported into the HC (Fig. 5e, f). Therefore, relatively small increases in river surface elevation that cause increased intrusion of RW and, in turn, a decrease in GW fraction may significantly alter spatiotemporal dynamics of DOC across the HC. This type of thresholding behavior is a critical feature to capture in process-based hydro-biogeochemical models of the HC.

**Microbial ecology and DOC biochemistry**. Despite stimulated heterotrophic respiration in response to RW intrusion, the phylogenetic composition of microbiomes across the HC responded weakly to the intrusion of RW (Supplementary Fig. 4). To determine turnover in microbiome composition we used the β-nearest taxon index (βNTI) metric, which quantifies the difference between observed compositional turnover and a stochastic expectation[37]. βNTI < − 2 indicates that ecological selection pressures are consistent through time/space (termed homogenous selection), resulting in little variation in microbiome composition. βNTI > + 2 indicates that ecological selection pressures change through time/space (termed variable selection), causing large shifts in microbiome composition. βNTI values between − 2 and + 2 indicate that the selective environment does not strongly determine differences (or similarities) between a given pair of microbiomes[38,39]. βNTI analysis revealed that microbiomes within each compartment of the HC system—river, nearshore hyporheic zone, or the unconfined GW aquifer—were governed by homogeneous selection (Supplementary Fig. 5).

Although HC microbiomes did not respond strongly to RW intrusion, a striking connection between microbiome composition and DOC composition was revealed when river microbiomes were compared with hyporheic zone microbiomes. Specifically, βNTI increased nonlinearly (to values > + 2) with increasingly large shifts in DOC composition (Fig. 6a), determined by

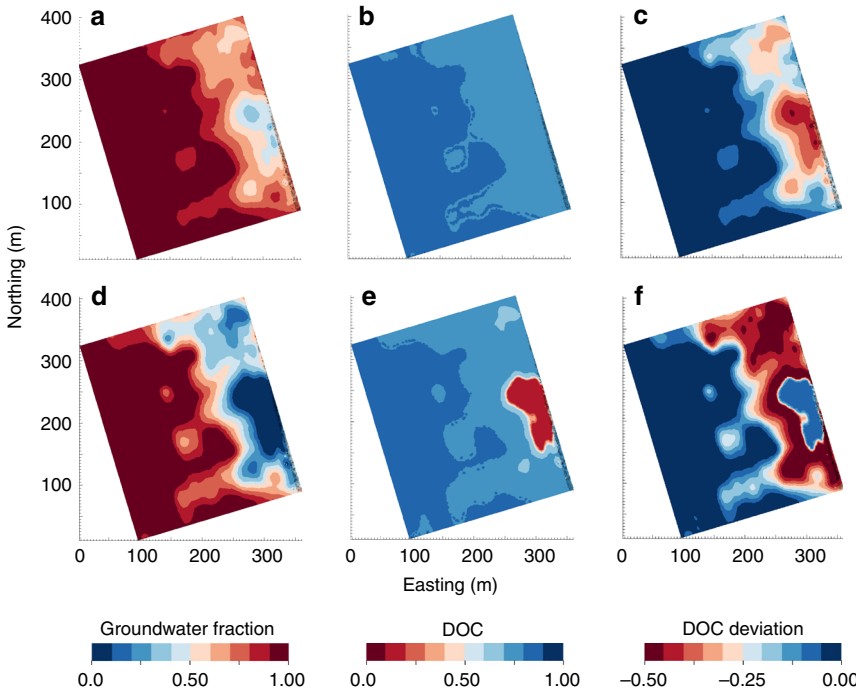

**Fig. 5** Spatial projections based on combining ERT estimates of groundwater fraction with aqueous chemistry. Spatial projections were generated across most of the domain that was sampled using groundwater (GW) wells (Fig. 2 and Supplementary Figs. 2,3)[34]. Due to changes in river stage, the HC experienced a broad range of GW–river water mixing conditions during the sampling period[34]. Two temporal snapshots were selected that differed in the degree of mixing. **a–c** Projections during a point in time with modest river intrusion and higher GW fraction. **d–f** Projections during a point in time with increased river intrusion and lower GW fraction. **a**, **d** Projections of GW fraction where pure GW is indicated by 1 and pure river water (RW) is indicated by 0. **b**, **e** Projections of dissolved organic carbon (DOC) concentrations generated by combining **a** and **d** with the empirical relationship shown by the gray line in Fig. 3a; DOC is presented as a fraction of RW DOC concentration, as in Fig. 3a. Between **b** and **e** there is a relatively small decrease in the GW fraction (i.e., a small increase in the RW fraction, as shown by comparing **a** with **d**). Due to nonlinearity in the relationship between GW fraction and DOC (Fig. 3a), this small decrease in GW fraction led to a large increase in DOC concentrations within the hyporheic corridor (HC). **c**, **f** Projections of the DOC mixing model deviations. The projections were generated by combining the spatial projection of GW fraction (**a**, **d**) with the empirical relationship shown by the gray line in Fig. 3d. The panels show that a large fraction of river-derived DOC was not transported into the broader HC, but that a small decrease in the GW fraction (increase in RW fraction) resulted in significant quantities of river-derived DOC being transported into the HC

**FTICR–MS.** Turnover in DOC composition was quantified with the Sorensen dissimilarity metric, ranging from 0 to 1; a value of 1 being completely dissimilar, using the presence/absence of MS peaks. Large differences (i.e., $\beta NTI > + 2$) in microbiome composition between the river and hyporheic zone were associated with clear shifts in DOC composition (Van–Krevelen analysis[40], Fig. 6b). In particular, there was a shift from lignin-like compounds composed of only C, H, and O in RW (Fig. 6c) to amino sugar-like compounds composed of C, H, O, and N in HZ water (Fig. 6d). Lignin-like CHO compounds that were over-represented in the RW may be the result of allochthonous (i.e., terrestrial) C inputs to RW[41], whereas amino sugars over-represented in the hyporheic zone likely reflect microbial processing of fresh plant material and microbial biomass[42].

The strong positive relationship between $\beta NTI$ and the FTICR-MS-based Sorensen dissimilarity indicated that shifts in DOC speciation were associated with shifts in deterministic ecological selective pressures that govern the composition of microbiomes[39]. To reveal key biochemical interactions between the microbiomes and DOC, we identified biochemical transformations that were differentially represented between RW and the hyporheic zone. To do so we focused on FTICR–MS data from samples associated with $\beta NTI > + 2$ (Fig. 6a). We focused on these samples, because $\beta NTI > + 2$ indicates a shift in microbiome composition that is deterministically driven by a shift in environmental conditions[38]. Within these samples, we identified FTICR–MS peaks unique to the river and those unique to the hyporheic zone. In turn, we

leveraged the high mass accuracy of the FTICR–MS to estimate—within each set of unique peaks—the frequency of 82 unique biochemical transformations[43] (see Methods and Supplementary Datas 1, 2). This approach follows from Longnecker and Kujawinski[44], and is based on counting the number of times each transformation was inferred within each set of peaks.

For each transformation in each data set, we estimated its fractional contribution to the total number of observed transformations; resulting values are akin to relative abundances within each data set. The fractional contribution of each transformation in the RW was subtracted from its fractional contribution in the hyporheic zone. Resulting values $> 0$ indicate higher relative representation in the RW and values $< 0$ indicate higher relative representation in the hyporheic zone (Fig. 6e). This comparison was enabled by relatively similar numbers of peaks and transformations in the hyporheic zone and RW data sets. Before removing shared peaks, the hyporheic zone and river had 10,708 and 11,086 peaks, respectively, and 117,651 and 108,120 transformations, respectively. There were 4,813 peaks in common, leaving 5,895 and 6,273 peaks in the unique peak sets for the hyporheic zone and river, respectively. The unique peak sets were associated with 34,954 and 31,307 transformations in the hyporheic zone and river, respectively. Normalized counts and river-to-hyporheic zone differences in normalized counts for each transformation are provided for the analyses with and without shared peaks removed (see Supplementary Datas 1, 2). Below we focus on interpreting analyses when shared peaks were

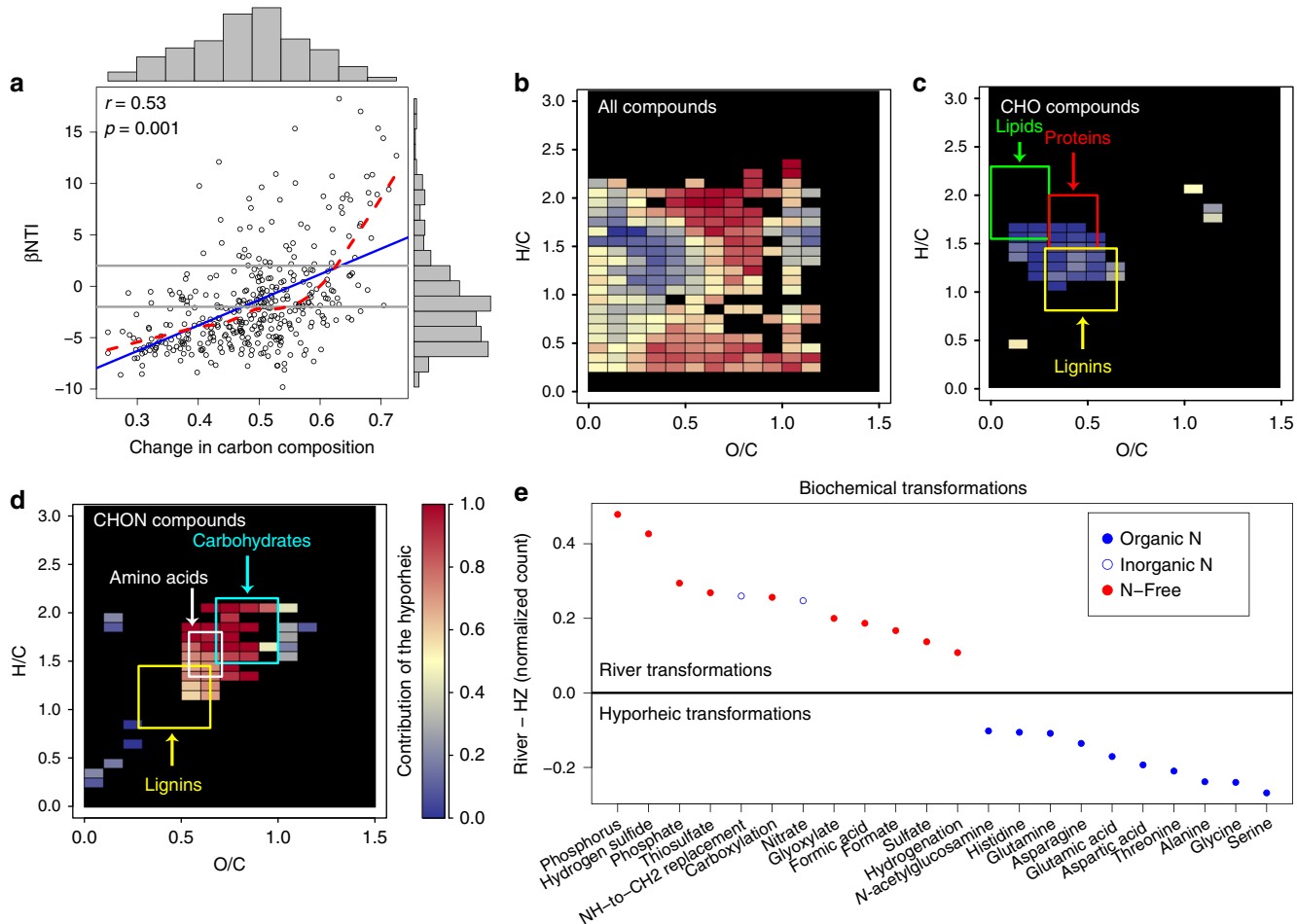

**Fig. 6** Microbial and organic carbon shifts between the river and HZ. **a** Ecological-null model outputs (βNTI) from comparisons between river and nearshore HZ microbiomes, as a function of changes in the organic carbon profile. Shifts in the carbon profile were quantified using the Sorensen dissimilarity metric (ranges from 0 to 1) on the presence/absence of peaks from the FTICR-MS data. **b**, **c** Van Krevelen diagrams based on FTICR-MS peaks unique to the RW or the HZ. Colors indicate the fraction of peaks found within a grid cell that were unique to the HZ (see color bar for scale). Overlaid rectangles indicate regions consistent with different classes of organic molecules. **b**–**d** differ in terms of showing data from all peaks (**b**), peaks with molecular formulas containing only C, H, and O (**c**), and peaks with molecular formulas containing only C, H, O, and N. **e** Biochemical transformations inferred from FTICR–MS data that appear to be overrepresented in either the river or the nearshore HZ; for clarity, only transformations with an absolute value on the vertical axis of 0.1 are shown (60 transformations not shown, see Supplementary Data 1). Values of 0 on the vertical axis indicate equal representation, whereas values above/below 0 indicate overrepresentation in the river/HZ, respectively. Colors indicate whether a given transformation involved N and closed vs. open blue circles indicate whether the N was organic or inorganic, respectively

removed, but our primary inferences were the same whether we kept or removed peaks shared between the hyporheic zone and river data sets (see below). We emphasize, however, that FTICR–MS is not quantitative in terms of concentrations of specific compounds. Our results should therefore be interpreted as providing new hypotheses about the interaction between HC microbiomes and DOC biochemistry.

The analysis of biochemical transformations revealed that deterministic shifts in microbiome composition were associated with a clear shift DOC biochemistry (Fig. 6e), providing mechanistic insight into the interactions between HC microbiomes and their biogeochemical environment. In particular, this analysis revealed a clear transition from N-free biochemical transformations in RW to N-containing transformations in the hyporheic zone that were largely associated with amino acids (Fig. 6e). This pattern was found both when removing or retaining FTICR–MS peaks shared between the hyporheic zone and river data sets (cf. Fig. 6e and Supplementary Fig. 6). Therefore, the ability to degrade protein may be a key functional trait selected for in hyporheic zone microbiomes. Furthermore,

ecological selective pressures in the RW appear to be consistent through time (Supplementary Fig. 5; also see Graham et al.[45]) and may not include selection for microbial taxa with proteolytic abilities. Although our results do not reveal what imposes selection within RW, they indicate that selection for proteolytic activity is strong within the hyporheic zone and is not a selective pressure in RW. The lack of selection for proteolytic activity in the RW microbiome therefore appears to be a key factor differentiating microbiomes between RW and the hyporheic zone. In our study system we have repeatedly found distinct microbiomes between RW and the hyporheic zone[1,15,45]. We expect this is generally true due to sustained biogeochemical differences between RW and hyporheic zone, even for systems with more temporal variation in selective pressures within RW. This hypothesis warrants broad evaluation.

Elevated proteolytic activity in the hyporheic zone may generally reflect higher cell densities and greater biogeochemical activity of the hyporheic zone, relative to RW[7]. Past work has, however, shown equivocal relationships between proteolytic activity and microbial biomass[46,47], instead finding that

proteolytic activity is induced under N or S limitation[47,48]. This suggests an important shift in limiting nutrients between RW and the hyporheic zone, and higher proteolytic activity in the hyporheic zone may specifically indicate N-limitation. Recent experimental work in the same field system, however, did not indicate N-limitation in the hyporheic zone; addition of N as nitrate did not elevate respiration rates, as measured by $CO_2$ production from sediment water slurries[49]. Furthermore, stable ecological selection pressures imposed on the RW microbiome (see above) suggest that the identity of limiting resources within RW is consistent through time. An intriguing question is whether the presence of dams in the study system leads to greater temporal stability in limiting resources, relative to undammed systems with greater temporal variability in river stage, flow velocities, and sediment transport. We look forward to evaluation of such questions and note that many inferences derived here should be treated as hypotheses that will need to be evaluated using additional controlled experiments and quantitative methods.

## Discussion

Integrating results across our analyses suggests a conceptual model that connects RW discharge dynamics, biogeochemistry, and microbiomes across the HC as follows: RW contains CHO lignin-like organic compounds—potentially indicating significant terrestrial carbon inputs to the RW—which are thermodynamically unfavorable for microbial oxidation, and these compounds enter the hyporheic zone following a rise in river stage. Once in the hyporheic zone, the CHO lignin-like compounds mix with more thermodynamically favorable GW DOC that, in turn, primes microorganisms to mineralize the river-derived DOC. This results in an increase in microbial activity, depletion of CHO lignin-like compounds and an increase in amino-acid-associated biochemical transformations in the hyporheic zone. These shifts in DOC composition are associated with a clear shift in deterministic ecological selection pressures that differentiate RW and hyporheic zone microbiomes, potentially due to selection for proteolytic activity in the hyporheic zone associated with N or S limitation. The influx of river-derived DOC into the hyporheic zone is followed by a rapid decline in DOC concentrations—due, in part, to microbe-driven DOC oxidation that is enhanced by GW–RW mixing. This occurs up to a GW fraction of 10% and across GW fractions larger than this threshold, DOC concentrations stabilize due to an energetic limitation imposed by low DOC concentration that persists despite accumulation of thermodynamically favorable DOC. The rapid decrease in DOC concentration as RW travels through the hyporheic zone results in minimal influences of RW intrusion on DOC concentrations and microbiome composition across the broader GW aquifer.

A key next step is to understand how shifts in DOC thermodynamics—arising from GW–RW mixing—impact microbial genomic potential and metabolic functioning across the HC. Given that mixing conditions can change rapidly (< 1 h) and frequently (daily)[50,51], we expect microbiomes across the HC to be well adapted to dynamic shifts in hydrologic and biogeochemical conditions[1,15,45], which is consistent with temporally stable selective pressures on microbiomes indicated by our βNTI analyses (Supplementary Fig. 5). In turn, genome-encoded metabolic potential may be relatively stable through time[52], whereas gene and protein expression patterns likely shift rapidly when the system moves across the critical mixing threshold of 10% GW. Understanding short- and long-term responses of microbial metabolism to mixing-induced changes in DOC thermodynamics will enable more process-based representation of the

feedbacks among hydrology, DOC biochemistry, and microbial ecology in hydro-biogeochemical models.

## Methods

**Field sampling.** Water samples were collected from the Columbia River and within the adjacent HC across a broad range of river stage conditions in 2013 (Fig. 2). GW was sampled with a submersible pump from 10 wells screened across the top of the subsurface aquifer. Hyporheic water and RW were sampled using a peristaltic pump from a piezometer installed into the riverbed to ~ 1 m depth and from an in-river location near the piezometer, respectively (Fig. 2). After purging pump lines a 0.22 µm polyethersulfone Sterivex filter (Millipore Co. Billerica, MA) was installed to filter water for aqueous chemistry samples. Filtered samples were collected into 40 ml borosilicate vials to be used for anions, DOC, and DIC, and stored at 4 °C. Three samples for FTICR–MS organic carbon analysis were also collected in 40 ml borosilicate vials and frozen at − 20 °C. After water sample collection, Sterivex filters were frozen on dry ice and stored at − 80 °C to be used for DNA extraction.

**Analytical methods.** DIC and non-purgeable organic carbon (referred to as DOC) were estimated with a Shimadzu combustion carbon analyzer TOC-Vcsh with ASI-V auto sampler. DIC was determined by injection into 25% phosphoric acid at ambient temperature with a calibration range of 0.35 to 40 mg l$^{-1}$ as C and both sodium bicarbonate solid and sodium carbonate solid standards (Nacalia Tesque). DOC was determined after acidification with 2 N HCl (five minute sparging) to remove DIC. The sample was then injected into the furnace at 680 °C with calibration from 0.35 to 3.5 mg l$^{-1}$ and limit of quantification of 0.20 mg l$^{-1}$ as C and potassium hydrogen phthalate solid standard (Nacalia Tesque).

A Dionex ICS-2000 anion chromatograph with AS40 auto sampler was used to determine nitrate and chloride concentrations (Guard column: IonPac AG18 guard, 4 × 50 mm; analytical column: IonPac AS18, 4 × 250 mm; suppressor: RFIC ASRS, 300 4 mm, self-regenerating; suppressor current: 99 mA). A 25 min gradient method was used with 25 µl injection volumes and a 1 ml min$^{-1}$ flow rate at 30 °C. The gradient consisted of 22 mM KOH (7 min), increasing KOH (1 min), 40 mM KOH (12 min), followed by decreasing KOH back to 22 mM (5 min). Aanion standards (1,000 mg l$^{-1}$; Spex CertiPrep, Metuchen, NJ) were diluted for a calibration range from 0.60 to 120 p.p.m., and measurement error was 2.5% for Cl$^-$ and 1.4% for $NO_3^-$.

**Organic carbon characterization.** Ultrahigh resolution characterization of DOC was done with a 15 Tesla Bruker SolariX FTICR–MS at the Environmental Molecular Sciences Laboratory (EMSL), a Department of Energy-Office of Biological and Environmental Research national user facility in Richland, WA. Samples were thawed overnight and then prepared by adding 0.5 ml of methanol (Optima LC/MS grade, Sigma Aldrich, Saint Louis, MO) to 0.5 ml of sample water in an Eppendorf 96 well plate fitted with 2 ml glass sleeves. Suwannee River Fulvic Acid standard (1 mg ml$^{-1}$ powder in MilliQ filtered water, diluted to 2 µg ml$^{-1}$, International Humic Substance Society) was used as control and injected every 15$^{th}$ sample to assure instrument stability. Lab blanks (1 : 1 by volume LC/MS grade methanol and MilliQ filtered water) were injected after the control to verify no sample carryover.

For the FTICR–MS analysis we analyzed three replicate samples for each sampling location on each sampling date. Samples were directly injected into the instrument using an automated custom-built system and ion accumulation time was optimized for all the samples. A standard Bruker electrospray ionization (ESI) source generated negatively charged molecular ions. Samples were introduced to the ESI source equipped with a fused silica tube (30 µm i.d.) through an Agilent 1200 series pump (Agilent Technologies) at a flow rate of 3.0 µl min$^{-1}$. Previous DOC characterization studies were used to set experimental conditions to optimal parameters (needle voltage, +4.4 kV; Q1 set to 50 m/z; heated resistively coated glass capillary operated at 180 °C). Individual scans (144) were averaged for each sample and internally calibrated using OM homologous series separated by 14 Da (–CH$_2$ groups). The mass measurement accuracy was typically within 1 p.p.m. for singly charged ions across a mass m/z range (100–1,100 m/z). The mass resolution was 350,000 at 339.112 Da. DataAnalysis software (BrukerDaltonics version 4.2) was used to convert raw spectra to a list of m/z values using FTMS peak picker (S/N threshold of 7; absolute intensity threshold of 100). To reduce cumulative errors, all sample peak lists for the entire dataset were aligned to each other before formula assignment to eliminate possible mass shifts that would impact formula assignment. In addition, the three replicates for each sampling location on each sampling date were collapsed into a single DOC profile by retaining all peaks observed across any of the three replicates. This was done to provide a more complete sampling of DOC species.

Putative chemical formulas were assigned using EMSL in-house software based on the Compound Identification Algorithm, described by Kujawinski and Behn[21], and modified by Minor et al.[22]. Chemical formulas were assigned based on the criteria of S/N > 7 and mass measurement error < 1 p.p.m., taking into consideration the presence of C, H, O, N, S, and P, and excluding other elements. Peaks with large mass ratios (m/z values > 500 Da) often have multiple possible candidate formulas; these peaks were assigned formulas through propagation of CH$_2$, O, and H$_2$ homologous series. To ensure consistent choice of molecular

formula when multiple formula candidates were found, the following rules were implemented: we consistently picked the formula with the lowest error with the lowest number of heteroatoms and the assignment of one phosphorus atom requires the presence of at least four oxygen atoms. The chemical character of thousands of features for each sample's ESI FTICR–MS spectrum was evaluated on van Krevelen diagrams. Compounds were plotted on the van Krevelen diagram on the basis of their molar H : C ratios ($y$ axis) and molar O : C ratios ($x$ axis). The Van Krevelen diagrams enabled comparison of the average properties of OM and the ability to assign DOC species to major biochemical classes, which included lipid-, protein-, lignin-, carbohydrate-, and condensed aromatic-like.

To characterize biochemical transformations that were potentially occurring within each sample, the mass differences between FTICR–MS peaks within each sample were compared with precise mass differences for commonly observed biochemical transformations[43]. A mass of 2.0156 Da, e.g., represents a hydrogenation/dehydrogenation reaction; a mass difference of 71.0371 indicates a reaction in which one alanine ($C_3H_5NO$) was lost. It is possible to infer biochemical transformations because the ultra-high mass accuracy of FTICR–MS. Within each sample we counted the number of times each transformation was observed based on precise mass differences between FTICR–MS peaks. Statistical analyses of the resulting transformation profiles are discussed below in sub-section 'Statistical analyses.'

We calculate the Gibbs free energy for the half reaction of organic carbon oxidation ($\Delta G^0_{Cox}$) by first estimating the nominal oxidation state of carbon (NOSC), per La Rowe and Van Cappellen[23]. NOSC is calculated from the equation:

$$NOSC = -((-Z + 4a + b - 3c - 2d + 5e - 2f)/a) + 4. \quad (1)$$

Here, $a$, $b$, $c$, $d$, $e$, and $f$ are the number of atoms of elements C, H, N, O, P, and S, respectively, in a given DOC species, and $Z$ is the net charge of the species. In turn, $\Delta G^0_{Cox}$ is estimated from the empirical equation:

$$\Delta G^0_{Cox} = 60.3 - 28.5(NOSC). \quad (2)$$

Values of $\Delta G^0_{Cox}$ are usually positive, indicating that the oxidation of DOC must be coupled to the reduction of a terminal electron acceptor. Importantly, a higher value of $\Delta G^0_{Cox}$ indicates a less thermodynamically favorable species[23]. The $\Delta G^0_{Cox}$ for a given sample was estimated as the median value; median was used instead of the mean due to some within-sample distributions of $\Delta G^0_{Cox}$ being skewed. It is noteworthy that each DOC species (i.e., each unique $m/z$ peak) was associated with a single $\Delta G^0_{Cox}$ value and DOC species were treated as present or absent. Peak intensities were not used due to uncertainty in how ionization efficiency of any given DOC species (and thus peak intensity) varied across samples due to changes in the profile of DOC species. As such, for a given sample, median $\Delta G^0_{Cox}$ was estimated by first calculating the $\Delta G^0_{Cox}$ for each peak in the sample and then finding the median value across all peaks within the sample. The resulting estimates for median $\Delta G^0_{Cox}$ were then analyzed by either regressing these values against GW fraction (Fig. 4a) or by grouping the samples based on their DOC concentration. To group samples, they were split into high and low DOC groups based on whether their DOC concentration was greater than or less than 0.4 mg l$^{-1}$, respectively. In turn, the distribution of median $\Delta G^0_{Cox}$ values within each group was summarized as a boxplot (Fig. 4b) and a $t$-test was used to evaluate whether the means of the two distributions were significantly different. The number of samples in the high and low DOC groups was 52 and 104, respectively, which does not pose a problem when using a $t$-test to compare means between the two groups. The number of peaks within each sample used to estimate the median $\Delta G^0_{Cox}$ varied from 211 to 3,499; sample-by-sample variation in the number of peaks is provided in Supplementary Data 3. Variation in the number of peaks across samples could result in samples with more peaks having more accurate estimates of median $\Delta G^0_{Cox}$, but this does not preclude comparing distributions of median $\Delta G^0_{Cox}$ values. This does, however, highlight the fact that methods do not currently exist to estimate the true level of uncertainty for within sample estimates of $\Delta G^0_{Cox}$. This suggests that the differences in $\Delta G^0_{Cox}$ we found between low and high DOC concentrations should be interpreted as generating a hypothesis that thermodynamic favorability of DOC varies with DOC concentration.

**DNA extraction and processing**. DNA was extracted from Sterviex filters using a modification of methods from Bostrom et al.[53]. Each Stervix filter housing was cracked open and the filter was removed using sterile instruments. Filter material was incubated at 85 °C for 15 min in lysis buffer. To avoid DNA fragmentation, the solution was slowly cooled and lysozyme was added to a final concentration of 1 mg ml$^{-1}$. The resulting solution was incubated at 37 °C for 30 min. SDS was added to 1% final concentration, and proteinase-K was added to 100 µl ml$^{-1}$ final concentration. The resulting solution was incubated at 55 °C for ~ 12 h. To facilitate cell lysis, the samples were subsequently exposed to three freeze–thaw cycles using liquid N$_2$ followed by warming to 55 °C. Samples were then vortexed, which was followed by isopropanol precipitation and pellet elution in Tris-EDTA (TE). The samples were treated with 10 µg ml$^{-1}$ RNase at 37 °C for 30 min, followed by phenol–chloroform clean-up, isopropanol precipitation, and elution in TE. Extracted DNA was stored at − 20 °C until shipment (on dry ice) to be sequenced using an Illumina MiSeq at the Environmental Sample Preparation and Sequencing

Facility at Argonne National Laboratory (ANL), Lemont, IL, following the same methods as in Stegen et al.[1].

Sequences generated by the ANL facility were processed in QIIME 1.8.0[54]. The function split_libraries_fastq.py was used to demultiplex the fastq-formatted sequences, with a Phred[55] quality cutoff of 20. To identify chimeras the function identify_chimeric_seqs.py was used to call USEARCH v6.1[56] with 'non_chimera_retention' set to 'intersection.' After removing identified chimeras (with function filter_fasta.py), the function pick_open_reference_otus.py was used to call USEARCH v6.1, pick operational taxonomic units (OTUs), and generate a phylogeny using SILVA (97% similarity, release 111) (http://www.arb-silva.de/) as the reference; default parameters were used except for the following modifications: 'suppress_de_novo_chimera_detection', 'suppress_reference_chimera_detection,' and 'derep_fullseq' were all set to 'True,' and 'prefilter_percent_id' was set to 0.

All additional analyses were carried out in R (http://cran.r-project.org/). OTUs identified as chloroplasts were removed and the OTU table was subsequently rarefied to 977 sequences—which resulted in one sample being dropped—using the rrarefy function in the vegan package. OTUs that had zero abundance across all samples, following rarefaction, were removed from both the OTU table and the phylogeny using the match.phylo.data function in the picante library. β-Mean nearest taxon distance (βMNTD) was used to quantify turnover in community phylogenetic structure; abundance weighted βMNTD was calculated with the comdistnt function in the picante package (see also refs [37,39,57,58]). To evaluate the potential for shifts in the relative balance between stochastic and deterministic ecological processes, randomizations were used to generate a null distribution (999 randomizations) for each βMNTD estimate; βMNTD is a pairwise distance metric such that a null βMNTD distribution was generated for each pairwise community comparison. The βNTI measures the difference between observed βMNTD and the mean of the null distribution in units of SD (see also refs [37,39,57,58]):

$$\beta NTI = (\beta MNTD_{obs} - \overline{\beta MNTD_{null}})/sd(\beta MNTD_{null}) \quad (3)$$

**Statistical analyses**. As in Stegen et al.[1], we used Cl$^-$ as a conserved tracer of GW–RW mixing. Owing to the long time span across which samples were collected, solute concentrations changed in the RW. Fitting a single mixing model to the entire dataset was not therefore feasible. Instead, for each sampling event we normalized all reactive solute (i.e., NO$_3^-$, DOC, and DIC) concentrations to the RW concentration during that sampling event. In this case, the normalized concentration of each solute in RW always takes on a value of 1 (e.g., in Fig. 3a the RW values are all exactly at 1), and values in all other samples are proportional to the normalized value. For example, a DOC concentration that was 50% of the RW concentration during a given sampling event has a value of 0.5 in Fig. 3a and a NO$_3^-$ concentration that was 20 times the RW concentration has a value of 20 in Fig. 3c; GW had a much higher NO$_3^-$ concentration than RW.

For each sample we converted Cl$^-$ concentration into an estimate of 'groundwater fraction.' This was done to facilitate the end-member mixing model analysis and to connect the reactive solute concentrations to the time lapse ERT images. GW fraction was estimated by first defining any Cl$^-$ concentration values over 17.5 mg l$^{-1}$ as pure GW; without any influence of RW, Cl$^-$ concentrations vary naturally across the GW aquifer we studied such that a threshold must be defined based on samples collected during low river stage, when there is no RW intrusion. We then found the mean Cl$^-$ concentration for all samples with Cl$^-$ > 17.5 mg l$^{-1}$. The resulting mean Cl$^-$ concentration was considered 100% GW. This value was used in the mixing model analyses. As 100% GW coincides with a mean Cl$^-$ concentration, GW fraction can exceed 100% (Fig. 3). The mean Cl$^-$ concentration from all RW samples was used to indicate 0% GW. The concentration of each reactive solute in GW was estimated as the mean value (normalized within each sampling event by the associated RW concentration) across all samples with Cl$^-$ > 17.5 mg l$^{-1}$. These analyses therefore provided reactive solute concentrations for each end member, RW, and GW; those estimates were combined with the estimated GW fractions to first fit end-member mixing models and then compare those models with the data from across the RW-to-GW gradient (Fig. 3a, b, c).

During the mixing model analyses we noted two samples that were clear outliers from the first two sampling events. To be conservative, we removed the first two sampling events from the mixing model analyses.

To evaluate the degree to which changes in GW–RW mixing conditions influenced microbial community composition through deterministic ecological selection, we used distance matrix regression and a permutation-based Mantel test to relate βNTI to changes in in Cl$^-$ concentration. The Mantel test was used to account for non-independence among the pairwise distance measurements (i.e., βNTI and change in changes in in Cl$^-$ concentration are both pairwise measurements of difference or distance). As we were interested in the influences of GW–RW mixing conditions, we excluded RW microbial communities from this analysis.

To evaluate the potential for DOC speciation to influence microbial community composition, we performed an exploratory analysis relating βNTI to changes in the DOC speciation profile. Change in the DOC speciation profile between all pairwise sample comparisons was based on the shifts in the presence/absence of $m/z$ peaks from the FTICR-MS. Shift in the DOC profile was quantified with the Sorensen dissimilarity metric, a common ecological metric used to measure shifts in community composition (distance function in the ecodist R package). The analysis

revealed a strong positive relationship between βNTI and Sorensen dissimilarity when comparing RW samples with nearshore hyporheic zone samples (Fig. 6a). Such a relationship was not observed in other comparisons; river samples compared with GW aquifer samples, hyporheic zone samples compared with GW aquifer samples, or simultaneously comparing samples from all parts of the system.

The strong positive relationship between βNTI and the FTICR–MS-based Sorensen dissimilarity indicated that shifts in DOC speciation were associated with shifts in deterministic ecological selective pressures that govern the composition of microbial communities[39]. As DOC interacts with microbial communities through biochemical transformations (see sub-section 'Organic carbon characterization'), we identified transformations that were overrepresented in either RW or the hyporheic zone. To do so we focused on FTICR–MS data from samples associated with βNTI > +2 in the comparison between RW and hyporheic zone microbial communities, shown in Fig. 6a. We studied these samples, because βNTI > + 2 indicates a shift in microbial community composition that is deterministically driven by a shift in environmental conditions[38]. In turn, we identified a set of samples from the river and another set from the hyporheic zone, all of which were associated with βNTI > +2. We then identified the FTICR–MS peaks that were unique to each set (i.e., peaks unique to the river and those unique to the hyporheic zone, for samples associated with βNTI > +2). Within these sets of peaks we inferred biochemical transformations and tallied up the number of times each transformation was observed. To compare the RW transformations with the hyporheic zone transformations we normalized the transformation counts within each data set and then subtracted the normalized representation of each transformation in the RW from its representation in the hyporheic zone. Resulting values > 0 indicate higher relative representation in the RW and values < 0 indicate higher relative representation in the hyporheic zone (Fig. 6e).

**Imaging RW intrusion**. Three-dimensional time-lapse ERT was used to monitor stage-driven RW intrusion into the aquifer by imaging the resultant changes in subsurface EC. The spatial domain covered by ERT monitoring is shown in Fig. 2 and Supplementary Figure 1. Details of the analysis are provided in Johnson et al.[34] and summarized here. ERT utilizes an array of electrodes to inject electrical current into the subsurface and measure the resulting potential field. A single measurement uses four electrodes, two as the current source and sink electrodes, and two as the positive and negative potential electrodes. Many such measurements are strategically collected and tomographically inverted to provide an estimate, or image, of the bulk subsurface EC distribution that gave rise to the measurements. A major component of EC in porous media is the conductivity of the pore fluid. At the monitored field site the EC of RW was ~ 50% of the GW conductivity, enabling it to act as a contrasting agent for time-lapse ERT imaging. Using low-stage conditions (when the HC is occupied only by ground water) as baseline, decreases in EC from baseline were diagnostic of the presence of RW. Subsequently, when ERT data were repeatedly collected and tomographically inverted, three-dimensional images of RW intrusion into and retreat from the GW-saturated HC[34,35] were produced. To illustrate, Supplementary Fig. 1 shows the surface electrode array superimposed on the ERT computational mesh.

The array consisted of 352 electrodes, divided into 11 electrodes lines spaced ~ 25 m apart, with 10 m spacing between electrodes on a given line. A single ERT survey, which constitutes a one time-lapse ERT snapshot, consisted of 40,454 dipole–dipole measurements (at numerous dipole spacing's), and required ~ 6 h to collect. Data were collected continuously, providing four snapshots per day, for a 60-day period comprising the period of spring-time peak flows (1 April to 29 June 2013).

Supplementary Figure 2 shows a single snapshot of the change in EC caused by the presence of RW at the peak stage observed during the monitoring period. Larger decreases in conductivity (i.e., cooler colors) are diagnostic of higher RW concentration, whereas white indicates no change from baseline, or 100% GW. To estimate the relative concentrations of RW and GW at a given point in space and time within the HC, each time-lapse image was transformed using an end member analysis[36]. End member analysis uses the ERT estimated conductivity at two points —when the aquifer is saturated with GW ($EC_0$) or RW ($EC_{rw}$)—to identify the endpoint conditions for a two-point calibration. Based on the fact that the EC of the subsurface is a linear function of the pore water EC[59], the RW fraction at each image pixel is given by

$$F_{rw} = \frac{EC_t - EC_0}{EC_{rw} - EC_0}, \qquad (4)$$

where $EC_t$ is the ERT estimated conductivity at time $t$. The corresponding GW fraction is given by

$$F_{gw} = 1 - F_{rw}. \qquad (5)$$

For example, equations 4 and 5 were used to convert the distribution of EC show in Supplementary Fig. 2 into the distribution of $F_{gw}$ shown in Supplementary Fig. 3.

**Data availability**. Data are available from the corresponding author upon request.

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

## Acknowledgements

This research was supported by the US Department of Energy (DOE), Office of Biological and Environmental Research (BER), as part of Subsurface Biogeochemical Research Program's Scientific Focus Area (SFA) at the Pacific Northwest National Laboratory (PNNL). PNNL is operated for DOE by Battelle under contract DE-AC06-76RLO 1830. A portion of the research was performed at Environmental Molecular Science Laboratory User Facility. We thank Amy Goldman for conceptual feedback and help in editing the manuscript.

## Author contributions

J.C.S., T.J., J.K.F., M.J.W., A.E.K., W.C.N., E.V.A., D.W.K., E.B.G, and J.Z. conceptualized the study. J.C.S., T.J., M.J.W., E.V.A., D.W.K., W.B.C., R.K.C., S.J.F., C.T.R., and M.T. carried out the study. J.C.S. conducted the statistical analyses. T.J. contributed the ERT projections. J.C.S. drafted the manuscript and all authors contributed to the writing.

## Additional information

**Competing interests:** The authors declare no competing financial interests.

