## [Peer Review File · Nature Communications]

Reviewers' comments:

Reviewer #1 (Remarks to the Author):

This paper contains many really interesting ideas and I found it to be an intriguing paper to read. The authors integrate many datasets to propose new insights in the biogeochemistry of hyporheic zones and argue that these systems are underappreciated in the terrestrial carbon cycle. If the authors are correct, their new model of hyporheic biogeochemistry could explain carbon turnover in mixing zones. However, I am not an expert in some aspects of this paper, especially the ERT section, so I defer to other reviewers for critical analysis of those sections. I am able to comment on the FT-ICR MS data, its treatment and interpretation in the context of carbon biogeochemistry. The data was collected well and appropriately. The data treatment and interpretation was not routine, however, and so merits a few comments. First, the idea of using biochemical transformations to examine organic matter composition is not new to this group and I was somewhat surprised to see the omission of recent work by Longnecker (in *Rapid Communications in Mass Spectrometry*). This paper lays the framework for some of the counting of biochemical transformations described here and should be cited. Given that this method of measuring biochemical reaction abundance is not fully quantitative, the authors should provide a fuller description of the caveats associated with their conclusions. Although I recognize that this is a tough thing to measure with FT-ICR MS data, the authors should be clear that their data is a first step in generating hypotheses that are then tested with more appropriate data.

I find the data transformations to be extensive, raising concerns that they have over-simplified the datasets to the detriment of data interpretation. For example, generating Figure 4e requires many different steps and simplifications. First the authors remove all peaks that occur in both end-member samples, thus focusing on the peaks unique to each end-member. How many peaks does this remove? Is the number of unique peaks in RW similar to the number of unique peaks in GW? Second the authors count the number of biochemical transformations in each end-member and normalize these values to the total number of biochemical transformations. Was the total number of biochemical transformations similar between RW and GW? If not, this could affect the comparison of the relative transformations between the two end-members. Lastly, the authors subtract the relative abundance of transformations in GW from that in RW. How many transformations had a difference of zero, and so are not plotted in Figure 4e?

Similarly, I do not understand the steps involved in generating Figure 2b. What do the box and whiskers represent in each DOC concentration level? Are the numbers of peaks within the different samples approximately comparable so that comparing their medians is appropriate?

The language surrounding the discussion of labile and refractory organic matter on page 6 is also confusing. What do the authors mean by “the low DOC concentration provides too little energy to microbial cells”? How is this idea resolved with their contention that labile DOC is accumulating in GW environments (while overall DOC concentrations decrease – lines 120-122)? The argument here is central to the conclusions of the paper and so the language here should be clear and precise.

In summary, this is good paper in that it integrates many different datasets to offer new insights into the biogeochemistry of hyporheic corridors (or hyporheic zones?). However, the data treatments may be over-simplifying the data and so providing erroneous conclusions. In addition, the writing is not clear in some places, leading to confusion for the reader. It is possible that the authors could make sufficient text changes to address these concerns.

Reviewer #2 (Remarks to the Author):

The manuscript “Influences of Organic-C Speciation on Hyporheic Corridor Biogeochemistry and Microbial Ecology” uses a host of high-end techniques to test mechanisms by which mixing of waters in the hyporheic zone may stimulate biogeochemical processing.

Overall I find that the work is incredibly novel and timely. The coupling of physical, chemical, and biological processes in the hyporheic zone is at the forefront of the research.

However – my overall recommendation is that the short format of Nature Communications is not well-suited to this publication. While the techniques are impressive, the limited space in the publication and even the supplement do not allow the authors to fully present the methods and interpretations. I take as an example the electrical geophysical work that is presented. It is not clear to me from the manuscript nor supplementary material where these data were collected, with what instrument, what quadrupoles were measured (nor how many), the quality of the data via summaries of stacking or inverse measurement error, how the data were inverted and with what quality, etc. I believe this information is all contained in reference 15 from the SI. In short, this is an example of the limitations of this short format.

While it is admirable that the authors are bringing together so many methods, this manuscript is fully reliant on readers having read multiple other manuscripts to understand what was done.

Indeed, few readers will be familiar enough with all of these methods and how to interpret them that the limited space for explanation is a hindrance to this study.

As a result, I recommend that the authors focus this effort for a long-form journal where the details can be presented and the manuscript made stand-alone. To be clear - I am extremely excited about the work and results – the “stand alone” issue is my primary critique.

Reviewer #3 (Remarks to the Author):

Stegen et al., present a novel study examining the role of DOC thermodynamic quality and quantity on hyporheic corridor (HC) biogeochemistry and microbial ecology. The authors used water samples collected from groundwater wells and a near shore piezometer along a GW-HC-RW continuum along the Columbia River at various river stages coupled to measurements of chloride, DIC, nitrate and DOC concentrations and speciation (FT-ICR-MS), ERT and mixing models to show that heterotrophic respiration was enhanced with GW-RW mixing but only along a narrow mixing of conditions (~0-10% GW). The authors suggest that this threshold was due to the fact that even though GW has a higher DOC energetic quality, the lower DOC concentrations prevents its use due to the dilution effect whereby it is too energetically costly for microbial uptake and degradation. Consequently, when there is a high level of RW mixing (i.e., 0-10% GW) which has a lower energetic quality but higher DOC, the microbial community in the HC is primed to oxidize the less energetically favourable DOC. Although I am less familiar with molecular microbiology techniques, the authors show compelling data from nearshore HZ and river biomes which shows a strong relationship between DOC composition and microbiome composition.

In terms of the bioenergetics aspects of this study, this study is novel and original in that the bioenergetic constraints on biogeochemical cycles in the HC remains largely unknown and untested. One of the main issues in the application and understanding of bioenergetics at the field scale is that it is largely theoretical because it is difficult to test in situ. This is also complicated by the fact that the theoretical foundation and the resulting microbial growth models by which bioenergetics is built upon is rooted in biotechnology which is focused on maximizing growth. Moreover, the majority of bioenergetics-based field studies are focused on deep sea environments (see Lever et al., 2015; Hoehler and Jørgensen, 2013; and references therein). Consequently, it is unclear as to whether or not studies and growth models developed for replete energy (i.e., biotechnology) conditions can be used for dynamic limited energy environments such as the HC. Therefore, the authors offer new insights on the role of GW-RW mixing on DOC dynamics in the HC. As the use and access to FT-ICR-MS becomes more mainstream I can see how this technique coupled with field measurements

(including ERT) and thermodynamic calculations may be implemented in reactive transport models to predict biogeochemical cycling in the HC. Moreover, there is also potential for this type of thermodynamic assessment (FT-ICR-MS + $[\Delta G]_{\text{Cox}}^{\circ}$ calc.) to be used in gene-centric biogeochemical models (e.g., Reed et al., 2014). Therefore, given the novelty and potential for the advancement of the understanding and modelling of HC biogeochemical cycling I recommend that the manuscript be accepted with minor revisions.

Comments:

Lines 70-73: Was pH measured? Were there differences in pH along the continuum? Can the difference in DIC be at least partially attributed to pH differences/geochemistry rather than respiration?

Lines 117-119: Is this referring to Fig.1a? I don't see a linear trajectory in 1a, are you referring to 1d? Please clarify and correct.

Line 187-190: It is not clear (to me at least) how to interpret these results. Are the biochemical transformations and the molecular transformations in Fig. 4e, the same thing? Are the molecular transformations listed on the X-axis Fig 4e just based on the presence or absence of these compound as identified by FT-ICR-MS or through the molecular data. Please clarify.

Lines 196-209: While the thermodynamic assessment is compelling, Fig. 1e shows that the microbial community in the HZ is potentially poised for proteolytic transformations. Was ammonium measured along the GW-HC-RW continuum? As suggested it is possible that the HC community is N-limited. Can the authors comment on whether or not the 0-10% GW threshold is a reflection of N-limitation rather than energy limitation? I think this needs to be clearer in the manuscript.

References:

Hoehler, T.M. and Jørgensen, B.B. (2013) Microbial life under extreme energy limitation. *Nature Reviews Microbiology* 11, 83-94.

Lever, M.A., Rogers, K.L., Lloyd, K.G., Overmann, J., Schink, B., Thauer, R.K., Hoehler, T.M. and Jørgensen, B.B. (2015) Life under extreme energy limitation: a synthesis of laboratory-and field-based investigations. *FEMS microbiology reviews* 39, 688-728.

Reed, D.C., Algar, C.K., Huber, J.A. and Dick, G.J. (2014) Gene-centric approach to integrating environmental genomics and biogeochemical models. *Proceedings of the National Academy of Sciences* 111, 1879-1884.

Reviewer #4 (Remarks to the Author):

Review of Stegen et al

Title: Influences of Organic-C Speciation on Hyporheic Corridor Biogeochemistry and Microbial Ecology

Summary: Stegen et al present a novel interpretation of the role of surface water (SW) and groundwater (GW) mixing in the hyporheic zone (HZ)/hyporheic corridor (HC). The authors suggest that thermodynamic favorability of DOC present in GW helps overcome limitations on consumption/decomposition of less thermodynamically favorable DOC present in surface water, and that this effect is facilitated by the mixing of SW and GW in the HZ. The authors also discuss this phenomenon in the context of broader impacts on biogeochemistry and microbial ecology (i.e. community composition) in the HZ.

This work is interesting and novel, the methods are sound and well performed, and their data interpretation provides unique insight on the role of surface and groundwater mixing in controlling the fate of DOC in river ecosystems. I recommend acceptance with minor revision.

My main critique is that a specific mechanism by which low concentrations of GW DOC can “prime” the oxidation of higher concentrations of lower quality DOC in the RW is not provided or hypothesized. Can the authors propose a mechanism that could explain these observations beyond the fact that mixing of SW and GW stimulates the priming?

Approach: Spatial and temporal modeling of changes in DOC and electrical conductivity are used to illustrate tipping points of mixing of DOC from SW and GW sources that results in stimulation of breakdown of less thermodynamically favorable DOC from SW sources. In addition, microbial

community measures were employed to illustrate that the microbial community shifts along with the DOC consumption patterns.

Review: The primary thrust of the authors hypothesis is that low concentration, but labile, DOC in GW can stimulate the breakdown/oxidation of low quality but relatively higher concentration DOC in SW when the two DOC sources are mixed in the HZ. The authors hypothesize that this stimulation proceeds via a “priming” effect that is catalyzed by a shift in the HZ microbial community. The proposed conceptual model the authors provide is compelling.

However, a mechanism by which low concentrations of high quality DOC in the HC can stimulate/prime the oxidation of relatively high concentrations of low quality DOC arising from RW intrusion is not provided.

Can the authors provide a hypothetical mechanism by which this pattern of DOC priming could occur, beyond the idea that mixing of RW and GW result in stimulation/priming of DOC oxidation? What about a low concentration DOC source would stimulate break down of low quality DOC? More typically, the process of “priming” is thought to be driven the the introduction (or release) of relatively large quantities of labile C that support microbial metabolism and growth. The release of high concentrations of labile C can result in increased respiration and potentially outgrowth of heterotrophic communities, followed by a corresponding net increase in metabolic demand. The additional demand then leads to microbial communities accessing/utilizing less labile C to sustain the metabolic demand. This can be driven by shifts in metabolism of existing community members or outgrowth of certain low DOC quality consuming populations or a combination of these events. Based on the DOC concentrations in GW as reported by the authors this mechanism may not be in play in the HC. Is there an alternate hypothetical mechanism that could drive the priming that was observed?

The authors provide some discussion of differential distributions of nutrient limitation in the RW and HC systems. Could the mechanism by which a low concentration of DOC can drive priming of a higher concentration of low quality DOC be driven by release from nutrient limitation instead of as the authors suggest a release of a thermodynamic limitation?

The authors used linear mixing models to establish expected changes in concentration of DOC, DIC, DO and Nitrate in the absence of stimulated respiration. Mixing models indicated a greater than expected decrease in DOC and DO and a corresponding greater than expected increase in DIC with respect to the linear model. Both measures indicate that the SW/GW mixing increased microbial respiration (but only at mixing levels < 10% GW). This approach to determining a priming effect is strong, however why the <10% GW limit? Is there something about the threshold of GW DOC that

puts a limit on the level of mixing tolerated to induce priming of surface water DOC? The authors attribute this to a thermodynamic limitation on priming. Does this imply a DOC concentration threshold on priming or a DOC-quality/quantity threshold? If so can the authors propose a $G_{Cox}:[DOC]$ threshold for a priming effect?

The application of FTICR-MS and microbiome composition to track interactions among differential sources of DOC and putative biochemical reactions catalyzed by the microbiome in response to the different suites of available C-substrates is quite compelling.

Particularly interesting is the shift from N-“free” to N containing metabolism associated with the shift from RW to HC microbiome-associated biochemical transformations. However, the authors note that “The ability to degrade protein may, therefore, be a key functional trait selected for in hyporheic zone microbiomes.” How can the ability to degrade protein by the microbiome as a key functional trait be disentangled from the degradation of protein present in hyporheic water and not RW? In other words, does the differential chemical signature in the DOC pools require that the HC microbiome have a particular functional character or does the presence of N-containing DOC in the HC water result in a shift in function by the microbial communities present without a shift in structure?

The following line of text in the manuscript may address the question above: “Furthermore, ecological selective pressures in the RW appear to be consistent through time (Supplementary Figure 4) and may not include selection for microbial taxa with proteolytic abilities.” In that the authors state that selective conditions in the RW appear to be relatively constant and not directed towards abilities associated with utilization of N-containing DOC. And as such RW DOC conditions may act as a source of stabilizing selection towards a particular RW microbiome functional character.

However, river systems are notorious for their high degrees of diel and inter-annual variability in DOC content and quality (i.e. labile vs. recalcitrant character and allochthonous vs. autochthonous inputs, etc.). Do the authors propose that relatively constant DOC conditions in the water column overlying a HC are necessary to drive the differences they observed in the microbiome structure and putative function? If so then would these observations be able to be extrapolated across broad spatial scales as the authors suggest or would they be limited to systems in which the selective conditions of the water column overlying a HC were relatively consistent?

The authors note that differential distributions of microbial functional potentials and community structures in the context of what they determined in terms of DOC quality elucidates a pattern of differential nutrient limitation in the RW and HC systems (N limited in RW vs. DOC quantity limited in

the HC). Similar to the question above, how applicable would these findings/patterns be in fluvial systems that may be less static in terms of DOC quantities and qualities?

Specific comments:

Lines 126-131 are redundant with the abstract that is embedded in the introduction to the manuscript.

#####

Reviewers' comments:

Reviewer #1 (Remarks to the Author):

This paper contains many really interesting ideas and I found it to be an intriguing paper to read. The authors integrate many datasets to propose new insights in the biogeochemistry of hyporheic zones and argue that these systems are underappreciated in the terrestrial carbon cycle. If the authors are correct, their new model of hyporheic biogeochemistry could explain carbon turnover in mixing zones. However, I am not an expert in some aspects of this paper, especially the ERT section, so I defer to other reviewers for critical analysis of those sections. I am able to comment on the FT-ICR MS data, its treatment and interpretation in the context of carbon biogeochemistry. The data was collected well and appropriately. The data treatment and interpretation was not routine, however, and so merits a few comments. First, the idea of using biochemical transformations to examine organic matter composition is not new to this group and I was somewhat surprised to see the omission of recent work by Longnecker (in Rapid Communications in Mass Spectrometry). This paper lays the framework for some of the counting of biochemical transformations described here and should be cited.

This paper is now cited and the following sentence has been added to the 3rd paragraph of the 'Microbial ecology and DOC biochemistry' subsection: "This approach follows from Longnecker and Kujawinski, and is based on counting the number of times each transformation was inferred within each set of peaks."

Given that this method of measuring biochemical reaction abundance is not fully quantitative, the authors should provide a fuller description of the caveats associated with their conclusions. Although I recognize that this is a tough thing to measure with FT-ICR MS data, the authors should be clear that their data is a first step in generating hypotheses that are then tested with more appropriate data.

We added caveat statements similar to those in Longnecker and Kujawinski (2016). In the 4th paragraph of the 'Microbial ecology and DOC biochemistry' subsection we added: "We emphasize, however, that FTICR-MS is not quantitative and our results should be interpreted as providing new hypotheses about the interaction between HC microbiomes and DOC biochemistry." In addition, in the last paragraph of the 'Microbial ecology and DOC biochemistry' subsection we added: "As noted above, however, these inferences should be treated as hypotheses that will need to be evaluated using controlled experiments and quantitative methods."

I find the data transformations to be extensive, raising concerns that they have over-simplified the datasets to the detriment of data interpretation. For example, generating Figure 4e requires many different steps and simplifications. First the authors remove all peaks that occur in both end-member samples, thus focusing on the peaks unique to each end-member. How many peaks does this remove? Is the number of unique peaks in RW similar to the number of unique peaks in GW?

Second the authors count the number of biochemical transformations in each end-member and normalize these values to the total number of biochemical transformations. Was the total number of biochemical transformations similar between RW and GW? If not, this could affect the comparison of the relative transformations between the two end-members.

The number of peaks and transformations found in the hyporheic zone and river were

similar before and after removing shared peaks, and inferences drawn from the analyses did not depend on whether shared peaks were removed. In the 4th paragraph of the 'Microbial ecology and DOC biochemistry' subsection we added the following text related to these points: "This comparison was enabled by relatively similar numbers of peaks and transformations in the hyporheic zone and river water datasets. Prior to removing shared peaks the hyporheic zone and river had 10708 and 11086 peaks, respectively, and 117651 and 108120 transformations, respectively. There were 4813 peaks in common, leaving 5895 and 6273 peaks in the unique peak sets for the hyporheic zone and river, respectively. The unique peak sets were associated with 34954 and 31307 transformations in the hyporheic zone and river, respectively. Normalized counts and river-to-hyporheic zone differences in normalized counts for each transformation are provided for the analyses with and without shared peaks removed (see Supplementary Tables 1 and 2). Below we focus on interpreting analyses when shared peaks were removed, but our primary inferences were robust to keeping or removing peaks shared between the hyporheic zone and river datasets (see below). We emphasize, however, that FTICR-MS is not quantitative in terms of concentrations of specific compounds. Our results should therefore be interpreted as providing new hypotheses about the interaction between HC microbiomes and DOC biochemistry."

In the second to last paragraph of the 'Microbial ecology and DOC biochemistry' subsection we reference a new supplementary figure based on retaining all peaks, which shows the same qualitative pattern as Figure 5e (previously Figure 4e) and leads to the same inferences. The added text is: "This pattern was found both when removing or retaining FTICR-MS peaks shared between the hyporheic zone and river datasets (cf. Fig. 5e, Supplementary Figure S7)."

Lastly, the authors subtract the relative abundance of transformations in GW from that in RW. How many transformations had a difference of zero, and so are not plotted in Figure 4e?

There were 60 transformations that had a difference near zero between the river and hyporheic zone counts, and were therefore not plotted in Figure 5e, for clarity. A similar approach was taken in the analysis that retained all peaks (see Supplementary Figure S7). For both approaches, tables are now included that provide normalized counts and differences for all transformations (see Supplementary Tables 1,2).

In addition, we further separated N-containing transformations into those involving organic vs. inorganic N. This difference is indicated in the new Supplementary Tables 1,2 and in Figure 5e and Supplementary Figure S7.

Similarly, I do not understand the steps involved in generating Figure 2b. What do the box and whiskers represent in each DOC concentration level? Are the numbers of peaks within the different samples approximately comparable so that comparing their medians is appropriate?

To clarify what was done to generate Figure 3b (previously Figure 2b) and to address the point about among-sample variation in number of peaks we added a significant amount of information to the last paragraph within the 'Organic carbon characterization' subsection within the Methods section (now part of the main manuscript). Sample to sample variation in the number of peaks used to estimate Gibbs free energy does not preclude comparing those estimates across samples. Instead, it indicates that samples with more assigned peaks will have estimates of Gibbs free energy that are closer to the real population median. This does highlight an important point, which is that methods do not currently exist to estimate the true level of uncertainty for within sample estimates of Gibbs free energy. This suggests that the differences we found between low and high DOC concentrations should be interpreted as generating a hypothesis that thermodynamic favorability of DOC varies with DOC concentration. We now include a caveat related to this point in the 'Organic carbon characterization' subsection: "This does, however, highlight the fact that methods do not currently exist to estimate the true level of uncertainty for within sample estimates of ΔG_{Cox}^0 ". We also provide the number of peaks in each sample in

a new supplementary table (Supplementary Table 3).

The language surrounding the discussion of labile and refractory organic matter on page 6 is also confusing. What do the authors mean by “the low DOC concentration provides too little energy to microbial cells”? How is this idea resolved with their contention that labile DOC is accumulating in GW environments (while overall DOC concentrations decrease – lines 120-122)? The argument here is central to the conclusions of the paper and so the language here should be clear and precise.

Thank you for pushing us to clarify these ideas. We made significant revisions to improve clarity in this section of text, which is the middle portion of the ‘Mixing models and DOC thermodynamics’ subsection. The text is not reproduced here because it spans more than 1 page.

In summary, this is good paper in that it integrates many different datasets to offer new insights into the biogeochemistry of hyporheic corridors (or hyporheic zones?). However, the data treatments may be over-simplifying the data and so providing erroneous conclusions. In addition, the writing is not clear in some places, leading to confusion for the reader. It is possible that the authors could make sufficient text changes to address these concerns.

Modifications made in response to the detailed comments above led to more transparency in the analyses and confirmation that our inferences do not change when underlying choices made during the data analysis are altered. We feel this has greatly improved the paper and thank the reviewer for the helpful suggestions.

Reviewer #2 (Remarks to the Author):

The manuscript “Influences of Organic-C Speciation on Hyporheic Corridor Biogeochemistry and Microbial Ecology” uses a host of high-end techniques to test mechanisms by which mixing of waters in the hyporheic zone may stimulate biogeochemical processing.

Overall I find that the work is incredibly novel and timely. The coupling of physical, chemical, and biological processes in the hyporheic zone is at the forefront of the research.

However – my overall recommendation is that the short format of Nature Communications is not well-suited to this publication. While the techniques are impressive, the limited space in the publication and even the supplement do not allow the authors to fully present the methods and interpretations. I take as an example the electrical geophysical work that is presented. It is not clear to me from the manuscript nor supplementary material where these data were collected, with what instrument, what quadrupoles were measured (nor how many), the quality of the data via summaries of stacking or inverse measurement error, how the data were inverted and with what quality, etc. I believe this information is all contained in reference 15 from the SI. In short, this is an example of the limitations of this short format.

While it is admirable that the authors are bringing together so many methods, this manuscript is fully reliant on readers having read multiple other manuscripts to understand what was done. Indeed, few readers will be familiar enough with all of these methods and how to interpret them that the limited space for explanation is a hindrance to this study.

As a result, I recommend that the authors focus this effort for a long-form journal where the details can be presented and the manuscript made stand-alone. To be clear - I am extremely excited about the work and results – the “stand alone” issue is my primary critique.

Thank you for the helpful suggestion. Fortunately, Nature Communications allows for longer format. We brought in more information on the geophysical methods, including locations and analysis approaches. Much of this (and all other methods associated with the study) has been included in the main manuscript. To further facilitate the paper being stand-alone, we added supplemental figures that are modifications of key figures from Johnson et al. (2015, WRR). In response to Reviewer #1 we also added more detail on our

data analysis methods/choices and underlying data related to FTICR-MS. These details are also included in the Methods section ('Organic carbon characterization' subsection) that is now part of the main manuscript and are supported by 3 new supplemental tables providing processed as well as raw data.

Reviewer #3 (Remarks to the Author):

Stegen et al., present a novel study examining the role of DOC thermodynamic quality and quantity on hyporheic corridor (HC) biogeochemistry and microbial ecology. The authors used water samples collected from groundwater wells and a near shore piezometer along a GW-HC-RW continuum along the Columbia River at various river stages coupled to measurements of chloride, DIC, nitrate and DOC concentrations and speciation (FT-ICR-MS), ERT and mixing models to show that heterotrophic respiration was enhanced with GW-RW mixing but only along a narrow mixing of conditions (~0-10% GW). The authors suggest that this threshold was due to the fact that even though GW has a higher DOC energetic quality, the lower DOC concentrations prevents its use due to the dilution effect whereby it is too energetically costly for microbial uptake and degradation. Consequently, when there is a high level of RW mixing (i.e., 0-10% GW) which has a lower energetic quality but higher DOC, the microbial community in the HC is primed to oxidize the less energetically favourable DOC. Although I am less familiar with molecular microbiology techniques, the authors show compelling data from nearshore HZ and river biomes which shows a strong relationship between DOC composition and microbiome composition.

In terms of the bioenergetics aspects of this study, this study is novel and original in that the bioenergetic constraints on biogeochemical cycles in the HC remains largely unknown and untested. One of the main issues in the application and understanding of bioenergetics at the field scale is that it is largely theoretical because it is difficult to test in situ. This is also complicated by the fact that the theoretical foundation and the resulting microbial growth models by which bioenergetics is built upon is rooted in biotechnology which is focused on maximizing growth. Moreover, the majority of bioenergetics-based field studies are focused on deep sea environments (see Lever et al., 2015; Hoehler and Jørgensen, 2013; and references therein). Consequently, it is unclear as to whether or not studies and growth models developed for replete energy (i.e., biotechnology) conditions can be used for dynamic limited energy environments such as the HC. Therefore, the authors offer new insights on the role of GW-RW mixing on DOC dynamics in the HC. As the use and access to FT-ICR-MS becomes more mainstream I can see how this technique coupled with field measurements (including ERT) and thermodynamic calculations may be implemented in reactive transport models to predict biogeochemical cycling in the HC. Moreover, there is also potential for this type of thermodynamic assessment (FT-ICR-MS + ΔG_{Cox} calc.) to be used in gene-centric biogeochemical models (e.g., Reed et al., 2014). Therefore, given the novelty and potential for the advancement of the understanding and modelling of HC biogeochemical cycling I recommend that the manuscript be accepted with minor revisions.

Comments:

Lines 70-73: Was pH measured? Were there differences in pH along the continuum? Can the difference in DIC be at least partially attributed to pH differences/geochemistry rather than respiration?

We attempted to measure pH, but had issues with the meter. In turn, there is a partial dataset that which is now included in the new Supplementary Table 3. We regressed DIC against pH, which showed a very weak relationship with $R^2 = 0.06$, suggesting that pH was not a primary driver of the observed geochemical patterns.

Lines 117-119: Is this referring to Fig.1a? I don't see a linear trajectory in 1a, are you referring to 1d? Please clarify and correct.

Yes, that is referring to Fig. 1d (now Fig. 2d). We clarified that in the manuscript.

Line 187-190: It is not clear (to me at least) how to interpret these results. Are the biochemical transformations and the molecular transformations in Fig. 4e, the same thing? Are the molecular transformations listed on the X-axis Fig 4e just based on the presence or absence of these compound as identified by FT-ICR-MS or through the molecular data. Please clarify.

Apologies for being inconsistent with the terminology. As the reviewer suggests, 'molecular' and 'biochemical' were being used interchangeably. We see how this is confusing, and have changed to only using 'biochemical' when referring to the transformations. The transformations listed across the X-axis in Fig. 4e (now Fig. 5e) are based solely on the FTICR-MS data. We moved the methods section associated with these analyses into the main manuscript to help clarify this point (see subsection 'Organic carbon characterization').

Lines 196-209: While the thermodynamic assessment is compelling, Fig. 1e shows that the microbial community in the HZ is potentially poised for proteolytic transformations. Was ammonium measured along the GW-HC-RW continuum? As suggested it is possible that the HC community is N-limited. Can the authors comment on whether or not the 0-10% GW threshold is a reflection of N-limitation rather than energy limitation? I think this needs to be clearer in the manuscript.

Ammonium was not measured because it is thought to be at consistently low concentrations within our field system due to mineral sorption. If that assumption is true, that could indicate N-limitation, which would certainly align with the inferred proteolytic activity of the hyporheic zone community. While N undoubtedly plays an important role in the system, we believe that the 10% GW threshold does not reflect N-limitation. This assertion is supported by two observations. First, biogeochemical activity appears to lowest when nitrate is most abundant (i.e., when there is more than 10% GW). Second, we just published a study conducted in the same field system in which we experimentally assessed the influence of adding nitrate (at the GW concentration) to natural river water (Goldman et al. 2017; DOI 10.5194/bg-14-4229-2017). Respiration rate (as measured by CO₂ production in 24 hour batch reactors with native sediment) did not respond to the addition of nitrate. This suggests that the supply of nitrate by groundwater does not impact short-term biogeochemical rates in this field system. We added text summarizing these points to the last paragraph of the 'Microbial ecology and DOC biochemistry' subsection: "This suggests an important shift in limiting nutrients between RW and the hyporheic zone, and higher proteolytic activity in the hyporheic zone may specifically indicate N-limitation. Recent experimental work in the same field system, however, did not indicate N-limitation in the hyporheic zone; addition of N as nitrate did not elevate respiration rates, as measured by CO₂ production from sediment-water slurries."

References:

- Hoehler, T.M. and Jørgensen, B.B. (2013) Microbial life under extreme energy limitation. *Nature Reviews Microbiology* 11, 83-94.
- Lever, M.A., Rogers, K.L., Lloyd, K.G., Overmann, J., Schink, B., Thauer, R.K., Hoehler, T.M. and Jørgensen, B.B. (2015) Life under extreme energy limitation: a synthesis of laboratory-and field-based investigations. *FEMS microbiology reviews* 39, 688-728.
- Reed, D.C., Algar, C.K., Huber, J.A. and Dick, G.J. (2014) Gene-centric approach to integrating environmental genomics and biogeochemical models. *Proceedings of the National Academy of Sciences* 111, 1879-1884.

Reviewer #4 (Remarks to the Author):

Review of Stegen et al

Title: Influences of Organic-C Speciation on Hyporheic Corridor Biogeochemistry and Microbial Ecology

Summary: Stegen et al present a novel interpretation of the role of surface water (SW) and groundwater (GW) mixing in the hyporheic zone (HZ)/hyporheic corridor (HC). The authors suggest that thermodynamic favorability of DOC present in GW helps overcome limitations on consumption/decomposition of less thermodynamically favorable DOC present in surface water, and that this effect is facilitated by the mixing of SW and GW in the HZ. The authors also discuss this phenomenon in the context of broader impacts on biogeochemistry and microbial ecology (i.e. community composition) in the HZ.

This work is interesting and novel, the methods are sound and well performed, and their data interpretation provides unique insight on the role of surface and groundwater mixing in controlling the fate of DOC in river ecosystems. I recommend acceptance with minor revision.

My main critique is that a specific mechanism by which low concentrations of GW DOC can "prime" the oxidation of higher concentrations of lower quality DOC in the RW is not provided or hypothesized. Can the authors propose a mechanism that could explain these observations beyond the fact that mixing of SW and GW stimulates the priming?

Approach: Spatial and temporal modeling of changes in DOC and electrical conductivity are used to illustrate tipping points of mixing of DOC from SW and GW sources that results in stimulation of breakdown of less thermodynamically favorable DOC from SW sources. In addition, microbial community measures were employed to illustrate that the microbial community shifts along with the DOC consumption patterns.

Review: The primary thrust of the authors hypothesis is that low concentration, but labile, DOC in GW can stimulate the breakdown/oxidation of low quality but relatively higher concentration DOC in SW when the two DOC sources are mixed in the HZ. The authors hypothesize that this stimulation proceeds via a "priming" effect that is catalyzed by a shift in the HZ microbial community. The proposed conceptual model the authors provide is compelling.

However, a mechanism by which low concentrations of high quality DOC in the HC can stimulate/prime the oxidation of relatively high concentrations of low quality DOC arising from RW intrusion is not provided.

Can the authors provide a hypothetical mechanism by which this pattern of DOC priming could occur, beyond the idea that mixing of RW and GW result in stimulation/priming of DOC oxidation? What about a low concentration DOC source would stimulate break down of low quality DOC? More typically, the process of "priming" is thought to be driven the the introduction (or release) of relatively large quantities of labile C that support microbial metabolism and growth. The release of high concentrations of labile C can result in increased respiration and potentially outgrowth of heterotrophic communities, followed by a corresponding net increase in metabolic demand. The additional demand then leads to microbial communities accessing/utilizing less labile C to sustain the metabolic demand. This can be driven by shifts in metabolism of existing community members or outgrowth of certain low DOC quality consuming populations or a combination of these events. Based on the DOC concentrations in GW as reported by the authors this mechanism may not be in play in the HC. Is there an alternate hypothetical mechanism that could drive the priming that was observed?

We thank the reviewer for pushing us to consider and propose a more mechanistic explanation. We note that our hypothesized explanation that is now included in the paper is purely speculative, but we agree with the reviewer that including some speculation here is useful as it provides a falsifiable target for follow-on experiments. We added a new paragraph at the end of the 'Mixing models and DOC thermodynamics' subsection summarizing our thinking: "While we can only speculate about mechanisms underlying this priming effect, we hypothesize that DOC thermodynamics may regulate expression and/or activity of enzymes used in the processing and oxidation of DOC. Under this hypothesis, inputs of low concentration and thermodynamically favorable DOC from GW act as a signal that results in increased enzyme expression or activity. The regulatory signal may be

associated with the amount of energy gained per oxidation event. If the mechanism is via enhanced enzyme activity, it would suggest positive feedback regulation similar to that shown for the stringent response in *Escherichia coli*. We further hypothesize that the thermodynamics-based signal only operates when DOC concentrations are above a threshold needed to offset the costs of producing enzymes used for DOC oxidation. As discussed above, this threshold appears to be ~0.35-0.4 mg/L across both our hyporheic corridor system and the deep sea. Although speculative, this regulation-based hypothesis provides a starting point for future experiments designed to reveal underlying mechanisms."

The authors provide some discussion of differential distributions of nutrient limitation in the RW and HC systems. Could the mechanism by which a low concentration of DOC can drive priming of a higher concentration of low quality DOC be driven by release from nutrient limitation instead of as the authors suggest a release of a thermodynamic limitation?

This is possible and is similar to an idea provided by Reviewer #3. However, our experimental results recently published in Goldman et al. (2017; DOI 10.5194/bg-14-4229-2017) did not support the idea that addition of GW-derived nitrate releases the microbial community from N-limitation. We now include a summary of this idea and the Goldman et al. results in the last paragraph of the 'Microbial ecology and DOC biochemistry' subsection: "This suggests an important shift in limiting nutrients between RW and the hyporheic zone, and higher proteolytic activity in the hyporheic zone may specifically indicate N-limitation. Recent experimental work in the same field system, however, did not indicate N-limitation in the hyporheic zone; addition of N as nitrate did not elevate respiration rates, as measured by CO₂ production from sediment-water slurries."

The authors used linear mixing models to establish expected changes in concentration of DOC, DIC, DO and Nitrate in the absence of stimulated respiration. Mixing models indicated a greater than expected decrease in DOC and DO and a corresponding greater than expected increase in DIC with respect to the linear model. Both measures indicate that the SW/GW mixing increased microbial respiration (but only at mixing levels < 10% GW). This approach to determining a priming effect is strong, however why the <10% GW limit? Is there something about the threshold of GW DOC that puts a limit on the level of mixing tolerated to induce priming of surface water DOC? The authors attribute this to a thermodynamic limitation on priming. Does this imply a DOC concentration threshold on priming or a DOC-quality/quantity threshold? If so can the authors propose a $G_{Cox}:[DOC]$ threshold for a priming effect?

We hypothesize that the 10% GW threshold is due to the DOC concentration falling below ~0.4 mg/L at GW contributions greater than 10%. This inference is now described more fully in the second half of the 'Mixing models and DOC thermodynamics' subsection. In the revised text we provide more detail on the inferred mechanistic basis for a concentration-based threshold. The revised text is not provided here because it spans more than 1 page.

The application of FTICR-MS and microbiome composition to track interactions among differential sources of DOC and putative biochemical reactions catalyzed by the microbiome in response to the different suites of available C-substrates is quite compelling.

Particularly interesting is the shift from N-"free" to N containing metabolism associated with the shift from RW to HC microbiome-associated biochemical transformations. However, the authors note that "The ability to degrade protein may, therefore, be a key functional trait selected for in hyporheic zone microbiomes." How can the ability to degrade protein by the microbiome as a key functional trait be disentangled from the degradation of protein present in hyporheic water and not RW? In other words, does the differential chemical signature in the DOC pools require that the HC microbiome have a particular functional character or does the presence of N-containing DOC in the HC water result in a shift in function by the microbial communities present without a shift in structure?

Our ecological null model analyses using the 16S rRNA gene sequence data indicate that

shifts in microbiome structure between RW and the hyporheic zone are due to changes in environmental conditions, as opposed to being due simply to the movement of taxa (see Fig. 5a). That is, the differences are deterministically set by differences in the environment. As such, we infer that the ecological selective pressures are distinct between RW and the hyporheic zone, and we propose that the ability to degrade protein is one metabolic capability (i.e., functional trait) that is selected for in the hyporheic zone. Our results therefore indicate that there is a shift in community structure, in addition to function.

The following line of text in the manuscript may address the question above: "Furthermore, ecological selective pressures in the RW appear to be consistent through time (Supplementary Figure 4) and may not include selection for microbial taxa with proteolytic abilities." In that the authors state that selective conditions in the RW appear to be relatively constant and not directed towards abilities associated with utilization of N-containing DOC. And as such RW DOC conditions may act as a source of stabilizing selection towards a particular RW microbiome functional character.

Our interpretation is that selective pressures appear to be relatively constant through time in the RW. Our results don't provide any clear evidence, however, of what aspect of the RW environment is imposing selection. It may be DOC character, but as you rightly note below, we shouldn't expect DOC character in the RW to be constant. As such, there may be another (more stable) aspect of the RW that imposes relatively consistent selection on the RW microbiome. To address these points in the manuscript, we now state that our results do not clearly indicate what feature(s) of the RW environment imposes selection. This new text is in the second-to-last paragraph in the 'Microbial ecology and DOC biochemistry' subsection: "Furthermore, ecological selective pressures in the RW appear to be consistent through time (Supplementary Figure S6) and may not include selection for microbial taxa with proteolytic abilities. While our results do not reveal what imposes selection within RW, they strongly suggest that selection for proteolytic activity is strong within the hyporheic zone and is not a selective pressure in RW. The lack of selection for proteolytic activity in the RW microbiome difference in ecological selective pressures therefore appears to be a key factor differentiating microbiomes between RW and the hyporheic zone."

However, river systems are notorious for their high degrees of diel and inter-annual variability in DOC content and quality (i.e. labile vs. recalcitrant character and allochthonous vs. autochthonous inputs, etc.). Do the authors propose that relatively constant DOC conditions in the water column overlying a HC are necessary to drive the differences they observed in the microbiome structure and putative function? If so then would these observations be able to be extrapolated across broad spatial scales as the authors suggest or would they be limited to systems in which the selective conditions of the water column overlying a HC were relatively consistent?

Our interpretation is that the difference between RW and hyporheic zone microbiomes is not dependent on temporally stable selective pressures within the RW. We propose that instead, the microbiome differences are due to the biogeochemical environments being consistently different between the RW and hyporheic zone (see also Graham et al. 2017; DOI 10.1111/1462-2920.13720). Even for systems in which selective pressures in RW vary more through time (relative to our system), we expect to observe large microbiome differences between RW and the hyporheic zone. This interpretation is now summarized in the second-to-last paragraph of the 'Microbial ecology and DOC biochemistry' subsection: "In our study system we have repeatedly found distinct microbiomes between RW and the hyporheic zone. We expect this is generally true due to sustained biogeochemical differences between RW and hyporheic zone, even for systems with more temporal variation in selective pressures within RW. This hypothesis warrants broad evaluation."

The authors note that differential distributions of microbial functional potentials and community structures in the context of what they determined in terms of DOC quality elucidates a pattern of differential nutrient limitation in the RW and HC systems (N limited in RW vs. DOC quantity limited in the HC). Similar to the question above, how applicable would these findings/patterns be in fluvial

systems that may be less static in terms of DOC quantities and qualities?

The identity of limiting resources in RW and the hyporheic zone may be more temporally variable in other systems. In particular, an interesting possibility is that the presence of dams in our system may lead to more temporal stability in the identity of limiting resources. In undammed systems there is usually more within-year variation in river stage, flow velocities and sediment transport. This points to a need for dammed vs. undammed comparisons. We now include a summary of this idea in the last paragraph of the 'Microbial ecology and DOC biochemistry' subsection: "Furthermore, stable ecological selection pressures imposed on the RW microbiome (see above) suggest that the identity of limiting resources within RW is consistent through time. An intriguing question is whether the presence of dams in the study system leads to greater temporal stability in limiting resources, relative to undammed systems with greater temporal variability river stage, flow velocities and sediment transport."

Specific comments:

Lines 126-131 are redundant with the abstract that is embedded in the introduction to the manuscript.

The information provided in these two sections does overlap as the reviewer suggests. Our approach to the paper's structure is to give the reader the main take home points as early as possible. The rest of the manuscript then describes how we arrived at those inferences and provides additional interpretation. We feel this structure is effective at communicating the main points of the paper. Coming back to key points later in the paper provides greater understanding of how specific inferences were made. As the reviewer points out, this does result in a few lines of text in different parts of the paper conveying similar information. However, when reading the second instance of the text, the reader has more context and understanding of the underlying analyses. As such, while the ideas conveyed are the same, the purpose and utility of the text is not redundant. We would therefore very prefer to retain both instances.

REVIEWERS' COMMENTS:

Reviewer #1 (Remarks to the Author):

The authors have amply addressed my concerns regarding their treatment of the FT-ICR MS data. I am impressed at their diligence in addressing these issues and appreciate their willingness to expand and modify their discussions.

My only remaining concern is that the grey line in a few sub-plots of Figure 2 is practically impossible to see when printed. Please make this more distinct to aid the reader.

Reviewer #2 (Remarks to the Author):

The revised manuscript satisfies my primary critique of now being stand-alone and presenting the full range of required information to understand what was collected and how it was processed.

Reviewer #3 (Remarks to the Author):

The authors addressed all of my concerns. I recommend accept with no changes.

Reviewer comments:

Reviewer #1 (Remarks to the Author):

The authors have amply addressed my concerns regarding their treatment of the FT-ICR MS data. I am impressed at their diligence in addressing these issues and appreciate their willingness to expand and modify their discussions.

My only remaining concern is that the grey line in a few sub-plots of Figure 2 is practically impossible to see when printed. Please make this more distinct to aid the reader.

This edit was made.

Reviewer #2 (Remarks to the Author):

The revised manuscript satisfies my primary critique of now being stand-alone and presenting the full range of required information to understand what was collected and how it was processed.

Reviewer #3 (Remarks to the Author):

The authors addressed all of my concerns. I recommend accept with no changes.